# Upper-Ocean Processes Controlling the Near-Surface Temperature in the Western Gulf of Mexico from a Multidecadal Numerical Simulation

**Yangxing Zheng [1,*] , Mark A. Bourassa [1,2] , Dmitry Dukhovskoy [1] and M. M. Ali [1,3]**

1 Center for Ocean-Atmospheric Prediction Studies, The Florida State University, Tallahassee, FL 32306, USA; bourassa@coaps.fsu.edu (M.A.B.); ddukhovskoy@coaps.fsu.edu (D.D.); mmali@coaps.fsu.edu (M.M.A.)
2 Department of Earth, Ocean and Atmospheric Science, The Florida State University, Tallahassee, FL 32306, USA
3 Andhra Pradesh State Disaster Management Authority, Kunchanapalle 522501, India
* Correspondence: yzheng@fsu.edu; Tel.: +1-850-644-1159

**Abstract:** The upper-ocean processes controlling the near-surface layer temperature in the western Gulf of Mexico (GOM) are examined by estimating the contributing terms in the heat equation based on a 54-year simulation of an eddy-resolving HYbrid Coordinate Ocean Model (HYCOM). An eddy-active region defined by large surface eddy kinetic energy, representing the Loop Current eddies (LCEs) primary trajectory region, is selected for analysis. Both observations and the simulation reveal that the mean net surface heat flux cools the northern GOM and warms the southern GOM. Mean horizontal heat advection contributes to an overall cooling in the eddy-active region. Mean vertical heat advection has a strong seasonal variability associated with the strong seasonal cycle of the mixed layer process: winters tend to have a strong downward heat advection in the eddy-active region and a strong upward heat advection in the rest of the western GOM, while summers tend to have a weak advective heat flux. The downwelling (upwelling) is primarily due to the dominant anticyclonic (cyclonic) wind stress curl. Mean eddy heat flux convergence contributes to the overall warming in the upper ocean of the western GOM. Diffusive flux is not small across the thermocline, and it is expected to have an insignificant influence on the near-surface temperature.

**Keywords:** upper-ocean process; ocean surface temperature; heat budget analysis; HYCOM; Loop Current eddies; western Gulf of Mexico

## 1. Introduction

The Gulf of Mexico (GOM) is a semi-enclosed, partially land-locked, intercontinental, marginal sea. Numerous studies have demonstrated a strong influence of the GOM's near-surface temperature on weather and the formation of tornados and severe thunderstorms in the southeast U.S. [1–5]. Results from other studies have revealed a relationship between the intensification or weakening of tropical storms in the GOM and the near-surface temperature under the storm track [6]. Climate has an essential effect on land ecosystems [7,8], and the changes in the GOM's near-surface temperature affect the climate over the adjacent lands (North and South America), influencing the ecosystems there. Thus, it is critically important to understand how the upper-ocean processes control the near-surface temperature in the GOM. The heat and energy budget of the GOM is related to the Loop Current (LC) system in the eastern Gulf and the warm core anticyclonic eddies irregularly detaching from the LC (commonly called Loop Current eddies, LCEs) [9]. Several studies have examined the influence of the LC and LCEs on the upper ocean temperature anomalies in relation to severe weather in the region [5].

The LC brings a large volume of warm water into the GOM from the Caribbean Sea through the Yucatan Channel, and these warm water masses play an important role in

the development of tropical cyclones and hurricanes [10–13]. The LCEs are observed to translate westward and remain for several months in the western GOM [14,15]. Cold core cyclonic eddies that are predominantly formed along the LC and LCE fronts (also called frontal eddies) are characterized by smaller spatial scales and shallower vertical signatures when compared to LCEs. Yet, some of these cyclonic eddies are large enough to cause negative anomalies in the upper ocean temperature on a spatial scale of 50–200 km [16].

We investigate what upper-ocean processes control the near-surface temperature in the western GOM (i.e., west of 88° W), with an emphasis on mesoscale eddies. The analysis assesses the role of individual terms in the heat advection–diffusion equation. Prior to the satellite era, investigators examined the heat budget of the GOM using sparsely available oceanographic and meteorological records, which limited their ability to conduct a full heat budget analysis [17–20]. For example, a heat budget analysis was performed over the GOM but computed the important heat advection terms due to oceanic motions (e.g., horizontal and vertical heat advection, eddy heat flux divergence) as a residual [20]. The results were compared to those estimated directly with an approximate method that was used by Emery [21]. Using the monthly analyses of satellite and in situ datasets that span periods up to 32 years, Vukovich [22] argued that warm core eddies (i.e., anticyclonic eddies) are responsible for mass and heat redistribution in the western GOM while cold core eddies (i.e., cyclonic eddies) are primarily responsible for mass and heat redistribution in the eastern GOM. Zavala-Hidalgo et al. [23] studied the seasonal variability of surface heat fluxes using the Comprehensive Ocean-Atmosphere Data Set (COADS) climatology, bulk formulae, and radiation estimations from satellite measurements, and found the mean surface heat flux into the GOM is 9 W m$^{-2}$ with an amplitude of 168 W m$^{-2}$ for the annual cycle. They further discussed the relative importance of heat advection and entrainment on sea surface temperature (SST) using a primitive-equations inhomogeneous layered numerical model forced by climatological wind stress and highlighted the important impact of entrainment on SST. Some previous studies [24,25] highlighted the important influence of winds during fall and winter on increasing the heat input into the GOM, the effective heat transport by LCEs, and heat redistribution to the western Gulf by wind-induced shelf currents. Another earlier study [26] found that eddy heat flux convergence dominates the lateral advective fluxes and correlates well with the SST anomalies in the GOM using global climate models.

These earlier studies were an attempt to interpret the effects of LCEs on the heat budget of the GOM using datasets and/or simple numerical models. However, since the datasets used were sparse in time, irregular, and coarse (horizontally and vertically), some assumptions were made for the computation of the heat budget which might have reduced the reliability of these estimates. Thus, the previous studies were not able to directly estimate the upper-ocean heat budget terms induced by oceanic motions because of insufficient ocean data. This study complements and expands the previous studies by investigating the long-term consequence of upper-ocean processes on the near-surface temperature of the GOM. The present study is different from previous studies in at least three ways: (1) we investigate the long-term (seasonal and longer) consequences of mesoscale eddies in comparison to other physical processes (e.g., horizontal and vertical heat advections) on the upper-ocean heat distribution of the western GOM over the entire periods that ocean simulation covers, rather than only over the periods when LC eddies are present; (2) we compute the major terms (e.g., heat advection by horizontal and vertical flows, eddy heat flux convergence) directly from an ocean simulation instead of using a residual method [20], and the ocean model used in this study is also different from a layer model used by Zavala-Hidalgo et al. [23]; (3) the estimation of each heat contribution term is based on the 5-day averaged data derived from a multidecadal free running simulation of the 1/25 GOM Hybrid Coordinate Ocean Model (HYCOM, Section 2.1) presented in several studies [27–29]. The model provides 54 years of high spatial (~4 km) and temporal (3-hourly) resolution ocean fields, which captures more fine scale features of the ocean state and allows one to estimate the heat budget terms more accurately.

The remainder of this paper is organized as follows. Section 2 includes the description of the models and datasets. Section 3 presents the methodology. Section 4 describes the temporally mean SST and eddy kinetic energy representing the robustness of LCEs from HYCOM compared to observations. Section 5 presents the map distribution of time-mean heat terms and compares heat terms averaged in the western GOM to identify the relative roles of each physical process. Section 6 discusses some caveats and summarizes the results.

## 2. Model and Data

### 2.1. HYCOM

The present study uses a 54-year ocean simulation with the 1/25 GOM HYCOM [30–33] presented in [27]. Model output fields are available at https://www.hycom.org/data/goml0pt04/expt-02pt2 (accessed on 17 March 2022). The 54-year GOM HYCOM experiment was validated in [27] and analyzed in several other studies [28–30]. The model realistically reproduces LC variability and LCE statistics, and accurately simulates upper ocean and deep ocean dynamics [29]. Here, we provide validation of the SST fields from the GOM HYCOM with a variety of SST observations. The GOM HYCOM is forced by the hourly fields of the Climate Forecast System Reanalysis (CFSR) from 1992 to 2009. This 18-year record of surface forcing is repeated three times (three cycles), producing a continuous 54-year model integration. The nonlocal K-profile Parameterization (KPP) [34,35] turbulence closure scheme is employed for vertical mixing. Vertical diffusivity is determined from three components of mixing: shear generated mixing (based on a gradient Richardson number), background internal wave generated mixing, and double diffusive mixing. Horizontal diffusion is small because most eddy-related mixing and even submesoscale (~10 km) mixing are resolved. Second-order flux-corrected transport is used for scalar horizontal advection and momentum advection. The ends of the surface forcing time series are blended to prevent shocks in forcing between cycles, in a way to mimic the stochastic nature of atmospheric forcing in the real world. The 5-day averaged fields derived from the model outputs were used in this study.

### 2.2. Datasets

In this study, SST, upper ocean temperature, surface heat fluxes, and surface currents from different sources of datasets are first used to evaluate the performance of the HYCOM. All observational datasets and ocean analysis products used in this study, including spatial/temporal coverage and spatial/temporal resolutions, are summarized in Table 1.

First, a wide variety of ocean temperature datasets (including SST) are used for comparison with HYCOM. These datasets include the Generalized Digital Environmental Model version 3 (GDEM3) [36] and the nighttime Advanced Very High Resolution Radiometer (AVHRR) [37] operational SST implemented at NOAA and NASA.

Second, three heat flux datasets are used to show how net heat flux influences the near-surface temperature. These datasets consist of the monthly mean OAFlux [38,39] and monthly mean NCEP Climate Forecast System Reanalysis data (CFSR) [40]. Surface shortwave and longwave radiation in OAFlux are derived from the International Satellite Cloud Climatology Project flux dataset (ISCCP-FD) estimates [41,42] that are available from 1 July 1983. ISCCP-FD surface shortwave and longwave radiation products over the period 1984–2005 are combined with OAFlux and CFSR turbulent heat fluxes to produce the net surface heat fluxes.

Third, two ocean current products from satellite remote sensing are used to assess the performance of HYCOM in simulating the realistic eddy activity in the GOM. The gridded ($1/3° × 1/3°$, Mercator grid) product of the Ocean Topography Experiment (TOPEX)/Poseidon, European Remote Sensing Satellite-1 (ERS-1) and ERS-2, and Jason-1 and Jason-2 sea surface heights and geostrophic currents (computed from absolute topography) is used to validate the HYCOM's ability to simulate eddy activity. This product was produced by Segment Sol Multimissions d'Altimétrie, d'Orbitographie et de Localisation Précise/Data Unification and Altimeter Combination System (SSALTO/DUACS)

and distributed by Archiving, Validation, and Interpretation of Satellite Oceanographic data (AVISO) [43] with support from the Centre National d'Études Spatiales CNES (http://www.aviso.oceanobs.com/duacs/, accessed on 17 March 2022). The daily dataset covering the period from 1 January 1993 to 31 December 2014 is used in this study. In addition, 5-day interval near real-time ocean surface currents in $1/3°$ resolution derived from satellite altimeter and scatterometer Ocean Surface Current Analyses—Real-time (OSCAR) [44] data for the period 1 January 1993–31 December 2014 are used to evaluate the capability of simulating surface eddy activity by HYCOM.

**Table 1.** List of observational datasets and ocean analysis products used in this study. Variables from AVHRR and OSCAR are at 5-day averages; variables from HYCOM are at daily averages; all other parameters are at monthly averages. More details about datasets are in Section 2.

| Variables | Dataset | Spatial Coverage/Grid Spacing (º) | Temporal Coverage | References |
|---|---|---|---|---|
| SST | AVHRR | Global ocean 9 km × 9 km | January 1985–December 2002 | [37] |
| Ocean temp | HYCOM | GOM 1/25 × 1/25-L40 | 54 years | [27] |
| | GDEM3 | Global ocean 0.25 × 0.25-L78 | climatology | [36] |
| Ocean velocity | HYCOM | Global ocean 1/25 × 1/25-L40 | 54 years | [27] |
| Ocean surf velocity | AVISO | Global ocean 1/3 × 1/3 | January 1993–December 2014 | [43] |
| | OSCAR | Global ocean 1/3 × 1/3 | January 1993–December 2014 | [44] |
| Sea surf height | HYCOM | GOM 1/25 × 1/25-L40 | January 1993–December 2012 | [27] |
| Net surf heat flux | OAFlux | Global ocean 1 × 1 | July 1983–December 2009 | [38] |
| Surf latent/sensible heat flux | HYCOM | GOM 1/25 × 1/25-L40 | 54 years | [27] |
| | CFSR | Global ocean 1 × 1 | January 1979–December 2009 | [40] |
| Net surf shortwave/longwave flux | ISCCP-FD | Global ocean 1 × 1 | July 1983–December 2007 | [41] |

## 3. Methodology

### 3.1. Reynolds Averaging Heat Equation

The contribution of the individual terms to the change of the upper-ocean heat content is examined by using the Reynolds average thermal energy equation, written as an energy (enthalpy) equation. The velocity and temperature are decomposed into the time-averaged and fluctuating components. The equation is integrated over the upper layer from depth $z_0$ (see Appendix A for the detailed derivation)

$$\rho C_p \int_{z_0}^{0} \frac{\partial \overline{T}}{\partial t} dz = -\rho C_p \left[ \int_{z_0}^{0} \left( \overline{u} \frac{\partial \overline{T}}{\partial x} + \overline{v} \frac{\partial \overline{T}}{\partial y} \right) dz + \int_{z_0}^{0} \left( \frac{\partial \langle \overline{u'T'} \rangle}{\partial x} + \frac{\partial \langle \overline{v'T'} \rangle}{\partial y} \right) dz \right.$$
$$+ \int_{z_0}^{0} \overline{w} \frac{\partial \overline{T}}{\partial z} dz + \int_{z_0}^{0} \frac{\partial \langle \overline{w'T'} \rangle}{\partial z} dz - \int_{z_0}^{0} \nabla_h \kappa_h \nabla_h \overline{T} dz$$
$$\left. - \int_{z_0}^{0} \frac{\partial}{\partial z} \kappa_v \frac{\partial \overline{T}}{\partial z} dz \right] + \int_{z_0}^{0} Q_{sol} \gamma(z) dz$$

(1)

where $Q_{sol}$ is the net downward solar radiation that is absorbed with depth described by $\gamma(z)$. $C_p$ is the specific heat of seawater at constant pressure, $\rho$ is the density of seawater, and $\overline{u}$, $\overline{v}$, and $\overline{w}$ are the time-mean horizontal and vertical velocity components, respectively. $\overline{T}$ is time-mean temperature. The primed $u'$, $v'$, $w'$, and $T'$ are deviations of velocity and temperature from their time-mean $\overline{u}$, $\overline{v}$, $\overline{w}$, and $\overline{T}$, respectively, with a timescale for averaging that is defined in Section 3.2. $\nabla_h \cdot \left( \overline{V'T'} \right)$ and $\frac{\partial \left( \overline{w'T'} \right)}{\partial z}$ are the horizontal and vertical divergence of the eddy heat flux, and $\kappa_h$ and $\kappa_v$ are the horizontal and vertical diffusion coefficients, respectively. The term on the left-hand side of the equation is the rate of change of the upper-ocean heat content. As expected, it is small when averaged over several years to comply with the conservation of average internal energy. For better

understanding of the terms regarding vertical heat advection and vertical eddy heat flux convergence (see Appendix A), Equation (1) can be rewritten as

$$
\begin{aligned}
C_p \int_{z_0}^{0} \frac{\partial \overline{T}}{\partial t} dz = \quad & Q_{net} - \rho C_p \left[ \int_{z_0}^{0} \left( \overline{V}_h \cdot \nabla_h \overline{T} + \nabla_h \cdot \left( \overline{V'T'} \right) + \kappa_h \nabla_h^2 \overline{T} \right) dz \right] \\
& - \rho C_p \left( \hat{T} - \overline{T}(z_0) \right) \overline{w}(z_0) - \rho C_p \left[ \left( \overline{w'T'} \right)_0 - \left( \overline{w'T'} \right)_{z_0} \right] \\
& - \rho C_p \kappa_v \left. \frac{\partial \overline{T}}{\partial z} \right|_{z_0}
\end{aligned}
\tag{2}
$$

where $Q_{net}$ is the net surface heat flux, $V$ is the horizontal component of ocean velocity (u, v), $\hat{T}$ is the depth-averaged temperature from $z_0$ to surface. Integrated over the upper-ocean layer, the terms on the right-hand side of the equation are the net surface heat flux, horizontal heat advection by time-mean fields, the horizontal convergence of the eddy heat flux, the horizontal diffusion, the vertical heat advection by time-mean field, the eddy heat flux convergence in the vertical, and vertical heat diffusions, respectively. The vertical integral of vertical heat advection can be described as the heat flux across the bottom surface $z_0$ due to the mean vertical velocity and the difference between the depth-averaged temperature and temperature at $z_0$. The vertical integral of the vertical eddy heat flux convergence is determined by the turbulent vertical heat flux at surface (z = 0) and at bottom surface $z_0$. Note that $w' = \frac{\partial \eta}{\partial t}$ at z = 0, where $\eta$ is sea surface height. Note that the vertical diffusive flux acting on the large vertical temperature gradients at depth $z_0$ may not necessarily be small in this study.

In this study, all the heat terms in Equation (2) were directly computed from the HYCOM output fields except the horizontal and vertical heat diffusion terms. Heat terms were compared to examine the relative importance of upper-ocean physical processes in near-surface temperature. These were collected into a residual to close the heat budget. When the depth of the analyzed upper layer is shallower than ~100 m (i.e., $z_0 > 100$ m), the residual also includes a small contribution of the shortwave radiation that can penetrate down to 100 m, the magnitude of which depends on many factors such as the wind stirring, the solar zenith angle, the magnitude of incident shortwave flux, the optical properties of the ocean, etc. However, the shortwave contribution at depths >50 m is very small because the shortwave absorption decreases exponentially with depth. In clear water, the e-folding depth for attenuation of light is ~50 m [45]. Hence, the residual term is largely determined by diffusion.

*3.2. Timescale for Reynolds Averaging*

To compute the $\nabla_h \cdot \left( \overline{V'T'} \right)$ term in Equation (2), the timescale of the mean has to be defined. In this study, the timescale is defined as an average time ($\tau$) required for a LCE to pass across a fixed location. Therefore, the timescale is estimated as $\tau = D_{LCE}/U_{LCE}$, where $D_{LCE}$ is the spatial scale of the LCEs and $U_{LCE}$ is a characteristic westward drift speed of the LCEs. The size of the LCEs typically ranges from 200 to 400 km [46] and the LCE's mean translation speed ranges from 2 to 5 km per day [12]; hence, the timescale $\tau$ is estimated in the order of 100 days, suggesting that seasonal (90 days) averaging is appropriate for our goals. This timescale is comparable to the timescale for the Reynolds decomposition used in a similar study [47] which analyzed the eddy nutrient fluxes in the California current. Findings in [26] also support our choice of a seasonal timescale for averaging. Thus, the velocity and temperature fields in Equation (1) are separated into seasonal mean fields and perturbation components associated with the mesoscale eddies. Note that the velocities in the GOM mesoscale eddies (up to 2 m s$^{-1}$) are much larger than the typical translation speed of an eddy (2–5 km day$^{-1}$, i.e., 2–6 cm s$^{-1}$) [9,12,22]. Thus, the averaging timescale is chosen so that the impact of slowly westward-moving warm LCEs on the near-surface temperature of a location is separated from the influences of other rapid processes (e.g., the advection by the LCE's rotational current). Since cyclones also affect the near-surface

temperature in the GOM, and their timescales are comparable to the timescales of LCEs, the analysis in this study through seasonal averaging includes the role of cyclones as well. Note that we use 5-day average fields (derived from instantaneous 3-hourly HYCOM output fields) to compute the turbulent components, while the timescales shorter than 5 days are not resolved.

### 3.3. Estimating Geostrophic and Ageostrophic Current and Vertical Velocity

To examine the upper-ocean physical process related to the mean current, we partition the 5-day averaged current into two parts: geostrophic ($V_{geo}$) and ageostrophic currents ($V_{ageo}$). Zonal ($u_{geo}$) and meridional ($v_{geo}$) components of the geostrophic current at a depth of $z_0$ are computed by integrating the equations downward from the surface to a depth of $z_0$:

$$u_{geo} = -\frac{1}{\rho f}\frac{\partial}{\partial y}\int_{z_0}^{0} g\rho dz + u_s \tag{3}$$

and

$$v_{geo} = \frac{1}{\rho f}\frac{\partial}{\partial x}\int_{z_0}^{0} g\rho dz + v_s \tag{4}$$

where $u_s$ and $v_s$ are the zonal and meridional components of the surface geostrophic current, respectively, which are derived from the sea surface height (SSH) from the HYCOM output using the equations:

$$u_s = -\frac{g}{f}\frac{\partial \eta}{\partial y} \tag{5}$$

and

$$v_s = \frac{g}{f}\frac{\partial \eta}{\partial x} \tag{6}$$

where $f = 2\Omega\sin\theta$, the Coriolis parameter ($\Omega = 7.292 \times 10^{-5}\,\text{s}^{-1}$ is rotation rate of earth, and $\theta$ is the latitude), $\eta$ is the SSH (in meters), and g is the acceleration due to gravity. Ageostrophic current was computed as the difference between the total current and geostrophic current.

The vertical velocity (w) is a diagnostic variable which was computed directly from the simulated horizontal velocity components in the HYCOM vertical layers [48]. The vertical velocity w at z = 0 was derived from the time derivative of SSH ($w_0 = \frac{\partial \eta}{\partial t}$). Vertical heat advection ($-\overline{w}\frac{\partial \overline{T}}{\partial z}$) and vertical convergence of eddy heat flux ($-\frac{\partial\left(\overline{w'T'}\right)}{\partial z}$) were then computed based on the w and T profiles.

### 3.4. Depth for Heat Budget Analysis

Since LCEs can extend down to a depth of 1 km [46], they will affect the water temperature down to a depth of 1 km. In this study, we focus on how near-surface temperature can be influenced by LCEs on long time scales, much longer than the time scale of an individual LCE, because it is well known that a LCE can significantly affect the local SST when it passes. The depth $z_0$ in Equation (1) is assumed to be deep enough for all the shortwave radiation to be absorbed within the layer, and it should be a layer that well represents SST variations. Thus, $z_0$ is usually the mixed layer depth. Since the mixed layer depth in the GOM is usually shallow in summer and can reach 150 m deep in winter [49–51], we have conducted a heat budget analysis for three constant layers: 0–50 m, 0–100 m, and 0–150 m. We compare the relative roles of heat-contributing terms in Equation (2) for the three layers in this study.

## 4. Model Validation

The model has been validated in study [27] in terms of the LC and LCE features in the GOM. Here, we further validate the model in terms of temporal mean SST and surface eddy kinetic energy. The latter will be used to define an eddy-active region, in which the

upper ocean processes are analyzed to identify their relative roles in the change of the near-surface temperature.

*4.1. Mean SST*

Because SST is a good indicator of the near-surface temperature and is easier to observe from satellites, it is useful to validate the HYCOM simulation by comparing the modeled SST with observations. Spatial distribution of long-term mean SST in the GOM simulated in HYCOM is displayed and compared to the observed SST from GDEM3 and AVHRR (Figure 1). Although the averaging periods are different, these independent sources of data and the HYCOM simulation reveal some similar features of mean state SST in the GOM. For example, there are cooler waters with sharp SST gradients on the northern outer margins of the wide continental shelves. HYCOM is able to capture the Yucatan upwelling, as seen in the observations. The cool water pool in the Campeche Bank (90–88° W, 21–22° N) is observed to be associated with persistent upwelling driven by along-coast winds throughout the year [52]. The SST in the Bay of Campeche is higher than in the northern part of the western Gulf. A permanent warm tongue over the region (88–84° W, 21–26° N) is related to the LC that transports warm waters from the Caribbean Sea via the Yucatan Channel between Mexico and Cuba. The discrepancies between the HYCOM and SST observations are also obvious. For example, the SST gradients in HYCOM in the LC region are not as strong as those in AVHRR and GDEM3. Moreover, the SST in HYCOM is generally warmer than the observations in the western GOM. The pattern of SST in the western and eastern GOM is different from the observations. A relatively weak SST gradient and a uniform northeast–southwest orientation of isotherms appear in the open ocean of the western GOM (96–90° W, 22–27° N). These features are closely associated with the upper-ocean physical processes, which will be discussed in Section 5 through an estimate of heat-contributing terms in the heat equation for the upper ocean.

*4.2. Surface Eddy Kinetic Energy*

Because we are interested in how mesoscale eddies affect upper-ocean temperature, we identify a region where mesoscale eddies are energetic. Here, we define such a region where the values of eddy kinetic energy (EKE) derived from AVISO surface geostrophic velocity exceed 200 cm$^2$ s$^{-2}$. A region defined in this way is consistent with the primary LCE trajectory region of the western GOM [53]. To reveal the performance of HYCOM in generating realistic eddies, we compare EKE from HYCOM with EKEs derived from AVISO surface geostrophic velocity for the period January 1993–December 2014 and derived from OSCAR for near-real-time ocean surface currents for the period January 1993–December 2014, which are derived from satellite altimeter and scatterometer data. For AVISO, EKE is defined as EKE $= \left( u'^2_{geo} + v'^2_{geo} \right)/2$, where $u'_{geo}$ and $v'_{geo}$ are departures of the 5-day-averaged geostrophic velocity from their seasonally averaged values. For OSCAR and HYCOM, EKE is defined as EKE $= \left( u'^2 + v'^2 \right)/2$, where $u'$ and $v'$ are deviations of the 5-day-averaged total horizontal velocity from their seasonally averaged values. The spatial resolution of AVISO and OSCAR data is $1/3° \times 1/3°$. The estimates of EKE for HYCOM are based on $1/25° \times 1/25°$ grids.

Figure 2 illustrates the map distribution of the time-mean surface EKE (cm$^2$ s$^{-2}$) derived from the HYCOM simulation compared to those derived from OSCAR and AVISO. The EKE value of 200 cm$^2$ s$^{-2}$ computing from AVISO is indicated by a solid black outline that delimits the primary position for realistic active eddies in the western GOM. The demarcated region based upon the critical EKE value of 200 cm$^2$ s$^{-2}$ is found to be consistent with the primary location of the realistic paths of LC eddies in the western GOM, as shown by Hamilton et al. [53]. It is expected that the strongest eddy activity appears in the LC region where eddies initially detach from the LC. The largest EKE can also be caused by the effects of LC intrusions during different seasons. The strength of eddies gradually decays as they migrate southwestward in the open region of the western GOM, extending to 96°W in the west end. This spatial pattern of EKE activity derived from the HYCOM is similar to that

from AVISO and OSCAR, except that EKE activity also appears strong in the southwestern GOM where the LCEs dissipate. The magnitudes of EKE in HYCOM are larger than AVISO and OSCAR in most of the western GOM (Figure 2d,e). The magnitudes are different partly due to the different spatial resolutions. In this study, we are more interested in eddy activity in the western GOM since its westward propagation may influence the near-surface layer temperature there. Thus, the region surrounded by the solid black outline is assumed to be the mean primary position of observed active eddies in the western GOM, defined herein as an eddy-active region. Because HYCOM has a sufficiently fine horizontal resolution and is capable of resolving mesoscale and smaller-scale eddies, the magnitude of EKE in HYCOM is greater than AVISO and OSCAR, particularly in the LC region. This is partly because of the relatively coarse resolution in AVISO and OSCAR, which averages the smaller-scale high-amplitude features captured by the high-resolution model. Additionally, the use of surface geostrophic velocity may be underestimated to surface EKE in AVISO. A recent analysis revealed that the deep EKE from this HYCOM's simulation is largely consistent with the mean deep EKE derived from floats [29]. Regardless of these differences in the magnitude of EKE, HYCOM is reasonably capable of simulating active eddies in the GOM [27]. The prior section showed that SST patterns are also reasonable. These findings support the use of HYCOM in this study.

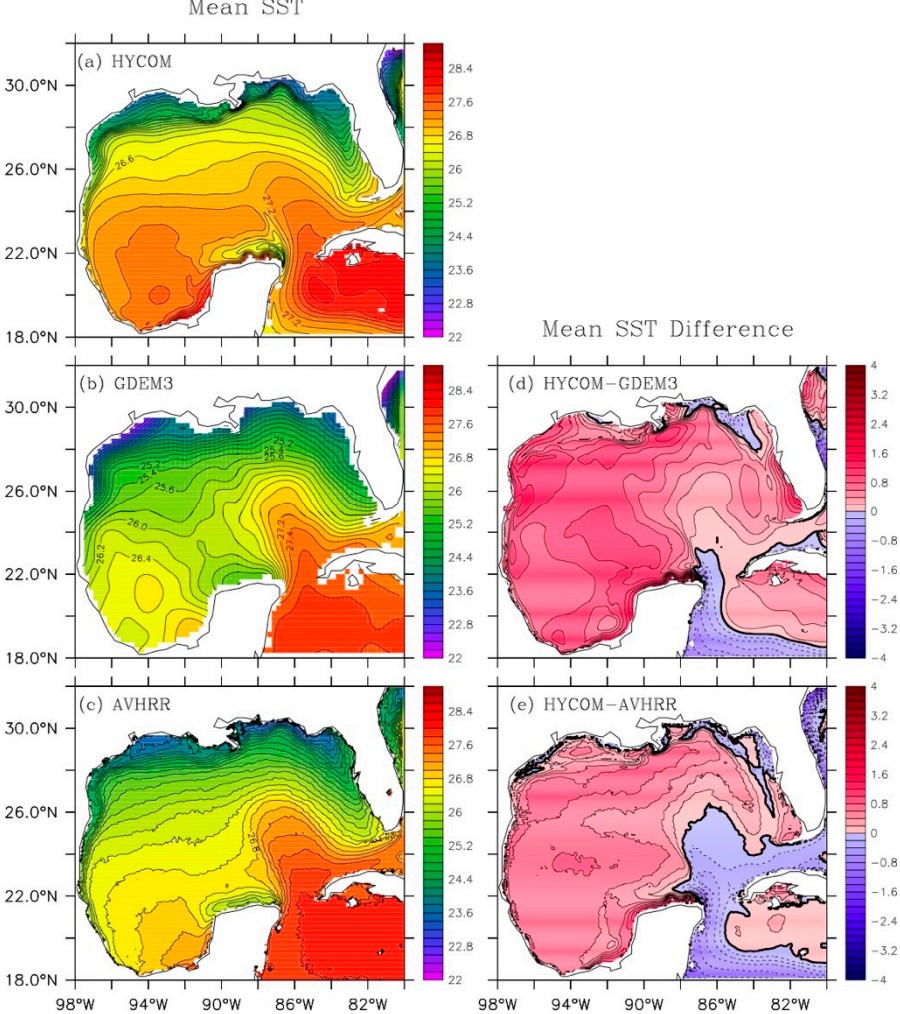

**Figure 1.** Spatial distribution of temporal mean SST (in °C) in the GOM obtained from (**a**) HYCOM, (**b**) GDEM3, (**c**) AVHRR and the difference of mean SST for (**d**) HYCOM–GDEM3, and (**e**) HYCOM–AVHRR. The averaging periods vary (see Section 2 and Table 1). Contour interval: 0.2 °C.

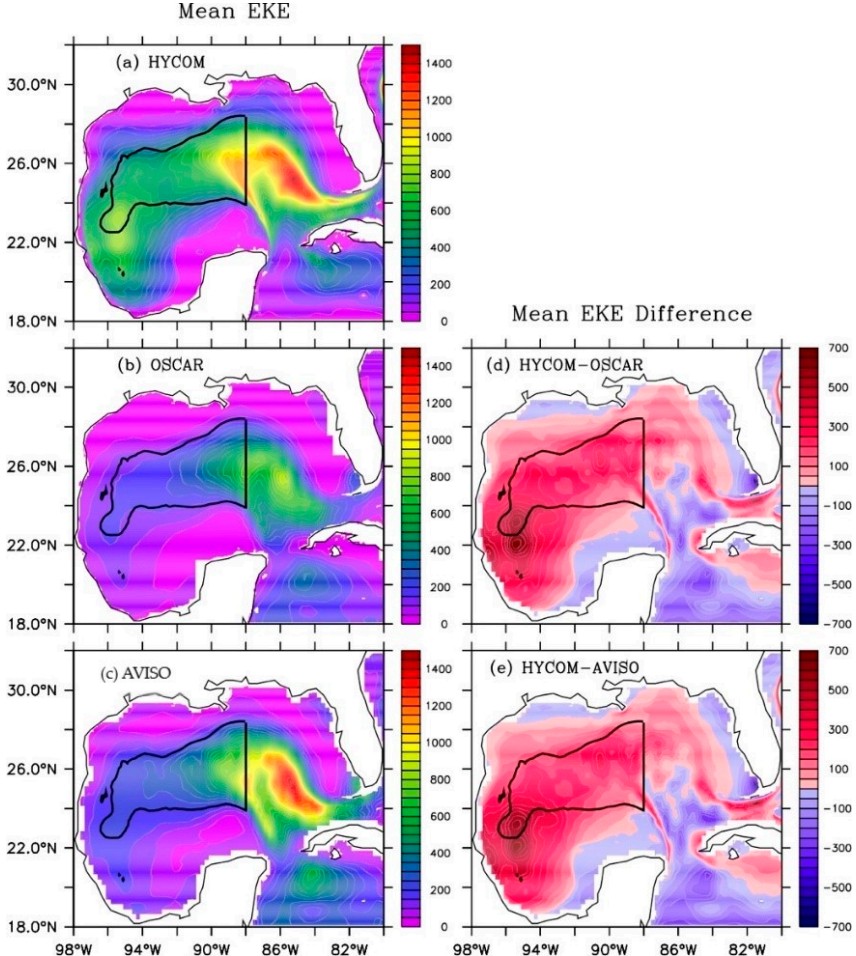

**Figure 2.** Maps of mean eddy activity captured by the temporal mean of surface EKE (cm² s⁻²) using total velocity for (**a**) HYCOM, (**b**) OSCAR, and geostrophic velocity for (**c**) AVISO, as well as the differences in surface EKE for (**d**) HYCOM–OSCAR and (**e**) HYCOM–AVISO. The bold black curve in (**a**–**e**) is a value of 200 cm² s⁻² derived from AVISO, which delimits the primary position of active eddies in the western GOM.

## 5. Results

### 5.1. Net Surface Heat Fluxes

Although the net heat flux term is not a major concern in this study, the time-mean net surface heat fluxes derived from the HYCOM simulation, and three other sources of datasets, are also compared. It is interesting to understand how the net surface heat flux contributes to the upper-ocean temperature change in the GOM. Figure 3 demonstrates the time-mean net surface heat fluxes from HYCOM, CFSR, and OAFlux, as well as the differences from CFSR and OAFlux.

The primary region of active eddies is delimited by the green curve defined in Section 4.2. In this study, we expect to find the common spatial features of net surface heat fluxes in the GOM from the HYCOM simulation and the three different datasets. Net surface shortwave and longwave fluxes from ISCCP-FD are used to compute the net surface heat flux for those datasets that only provide surface turbulent latent and sensible heat flux (e.g., OAFlux, NCEP R1, CFSR, etc.; see Table 1). Mean net surface heat fluxes in all datasets cause overall cooling in the northern portion of the GOM and overall warming in the southern part of the GOM. It is clear that more than half of the GOM, including the eddy-active region, is predominantly cooled via a strong air–sea interaction in HYCOM and the two datasets (NCEP R1 and CFSR). Differences in net surface heat flux among the comparison products and the HYCOM simulation are also obvious. For example, OAFlux

has positive fluxes in the Yucatan Channel and over the southwestern half of the GOM, while CFSR shows relatively weak negative flux in the eddy-active region. Net surface heat flux from HYCOM tends to cool the western GOM more compared to those from CFSR and OAFlux (Figure 3d,e). The negative surface heat flux in the eddy-active region is associated with warm LCEs travelling westward. Warm water transported from the Yucatan Channel is cooled by the net surface heat flux loss to the atmosphere with the latent heat flux being the primary contributor to the negative net surface heat flux. Both observations and the model simulation show that there is strong warming from the net surface heat flux in the southwestern corner of the GOM, the region of the maximum SSTs in the western GOM (as seen in Figure 1). The positive net surface heat flux is due to the small latent heat flux loss to the atmosphere when compared to those in the LC and deep-sea regions (not shown). Such cooling and warming of the ocean caused by the net surface heat flux must be balanced by upper-ocean processes, including horizontal and vertical heat advection, eddy heat flux convergence, diffusion, and dissipation. The major contributing terms, such as heat advection by mean flow and eddy heat flux convergence from eddies (perturbation terms), are computed directly from the HYCOM output. The diffusion terms for an upper-ocean layer are calculated as a residual if downward solar radiation is completely trapped with this layer.

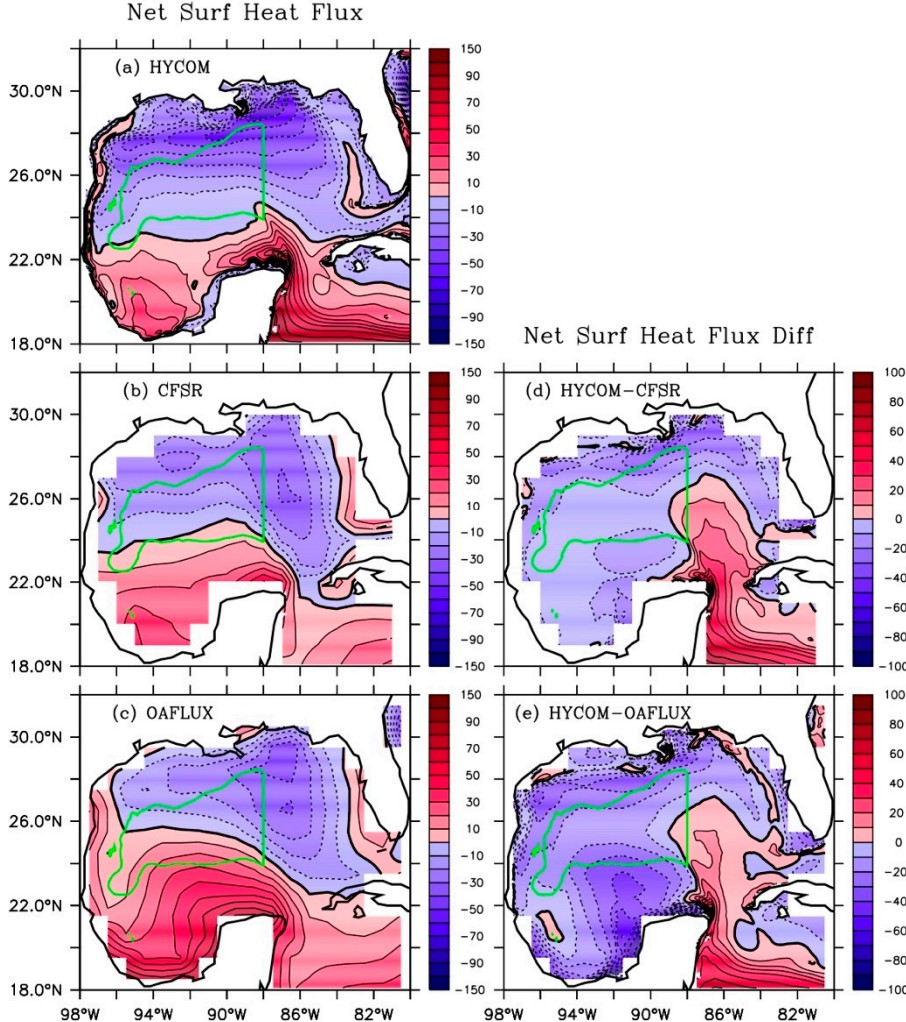

**Figure 3.** Horizontal structure of mean net surface heat fluxes (in W m$^{-2}$) in the GOM from (**a**) HY-COM, (**b**) CFSR, (**c**) OAFlux, and its differences for (**d**) HYCOM–CFSR and (**e**) HYCOM–OAFlux. Positive values (i.e., solid curves) denote warming the ocean. The primary position of active eddies in the western GOM is delimited by a green curve.

### 5.2. Horizontal Heat Advection

Horizontal heat advection is computed from the HYCOM output. Figure 4 shows the temporal mean horizontal heat advection in the upper 50 m, 100 m, and 150 m (Figure 4a–c), and the way in which the horizontal currents produce heat advection is revealed by the map of the time-mean horizontal currents and the time-mean temperature in HYCOM (Figure 4d–f). The primary location for active eddies is also denoted in each panel. Hereafter, any regions that are not deep enough to compute the heat terms in the corresponding depth layer will be omitted; thus, the heat terms in regions shallower than 50 m, 100 m, and 150 m are not shown in Figures 4a, 4b and 4c, respectively. Similarly, velocity and temperature are also omitted in such a region.

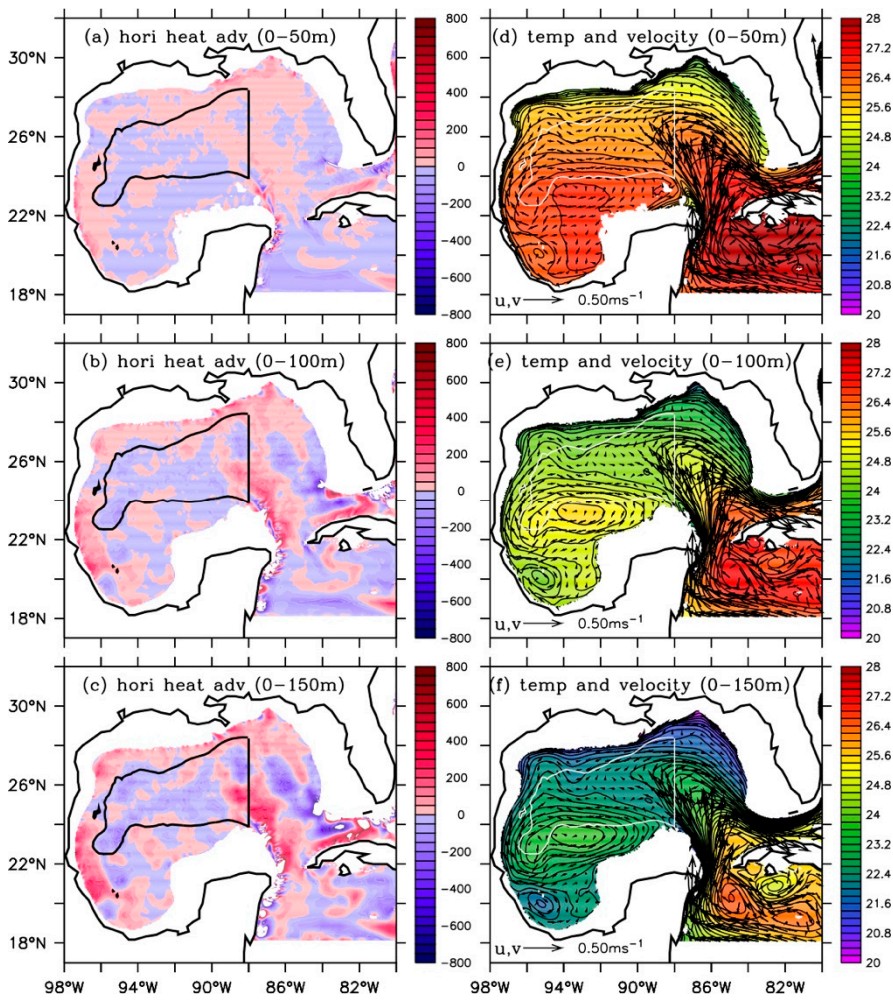

**Figure 4.** Mean horizontal heat advection (W m$^{-2}$) in the upper (**a**) 50 m, (**b**) 100 m, and (**c**) 150 m (**d**–**f**) are the mean horizontal currents (m s$^{-1}$) and temperature (°C) corresponding to (**a**–**c**). Positive values in (**a**–**c**) denote warming the ocean. The primary position of active eddies in the western GOM is also shown in the map delimited by black in (**a**–**c**) and by white curves in (**d**–**f**). In (**a**,**d**), regions deeper than 50 m are shown, while in (**b**,**e**), regions deeper than 100 m are shown, and in (**c**,**f**), regions deeper than 150 m are shown.

In contrast to the cooling over most of the northern GOM from the net surface heat flux, the mean horizontal advection in each layer appears similar and tends to cause either warming in regions adjacent to the coastline or cooling in most of the open ocean south of 26° N, where LCEs are generally active. The mean horizontal advection is fragmental in space. In the 0–100 m and 0–150 m layers, the cooling of the ocean is dominant in the primary region of the eddies. The relation between the mean horizontal currents and

the mean temperature overall explains these phenomena. The horizontal currents and mean temperature in the three different layers are similar. The weak warm advection near the U.S. coastal region is generally caused by the current moving northeastward from the warm offshore water to the cold shelf water. The warming in the western GOM near Mexico between 20° N and 24° N is caused by the northwestward current in a dominant anticyclonic circulation system that moves from warm offshore water to cold shelf water. The cooling in the open western Gulf is primarily induced by southeastward cross-isotherm flow in the northern part (including eddy-active region) and northwestward cross-isotherm flow in the southern part of an anticyclonic circulation. In the LC region, the strong northward currents via the Yucatan Channel are almost perpendicular to dense isotherms, causing a large warm advection in the north of Yucatan Channel. There is relatively weak horizontal advection in the Bay of Campeche because of the weak temperature gradients and the weak relationship between the mean currents and mean temperatures within the cyclonic circulation. Thus, the high SST in the Bay of Campeche (Figure 1a) is not largely caused by horizontal advection (Figure 4a), but by the net surface heat flux (Figure 3a).

We also computed the long-term mean horizontal heat advection caused by the geostrophic current (Figure 5) and ageostrophic current (Figure 6) in the above three layers.

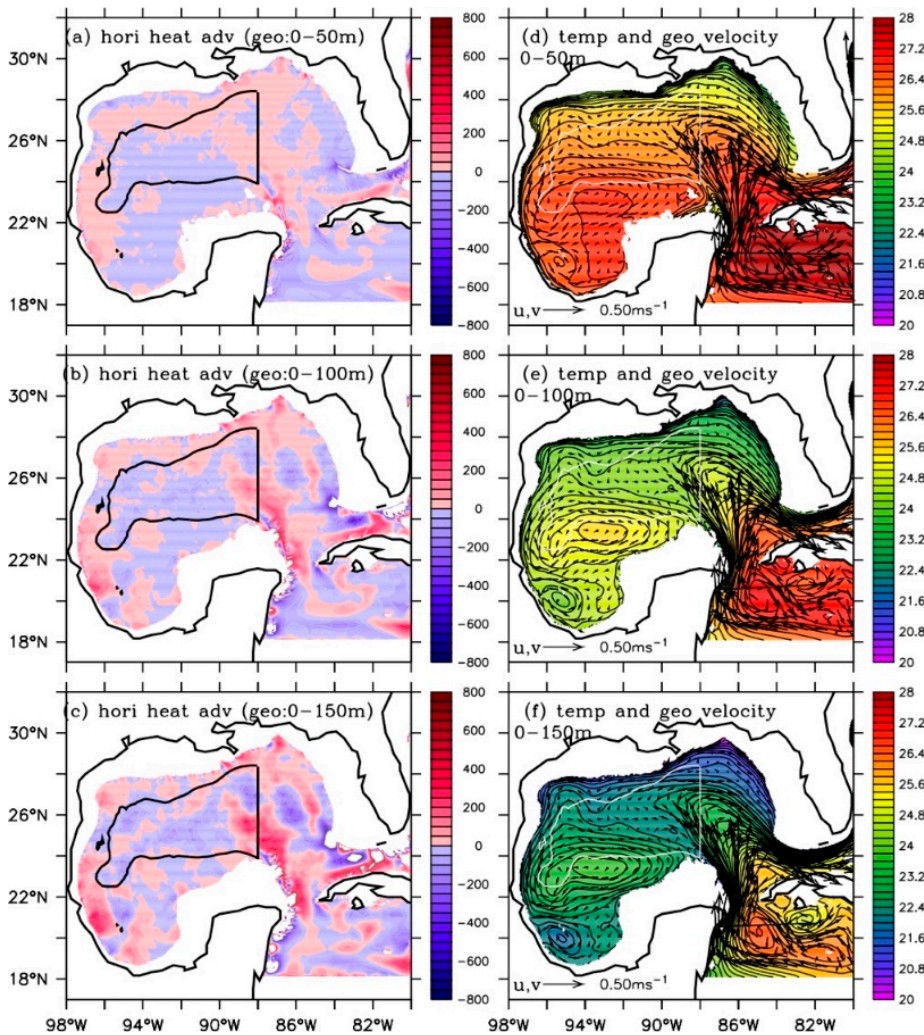

**Figure 5.** Mean horizontal heat advection caused by geostrophic current, arranged as in Figure 4.

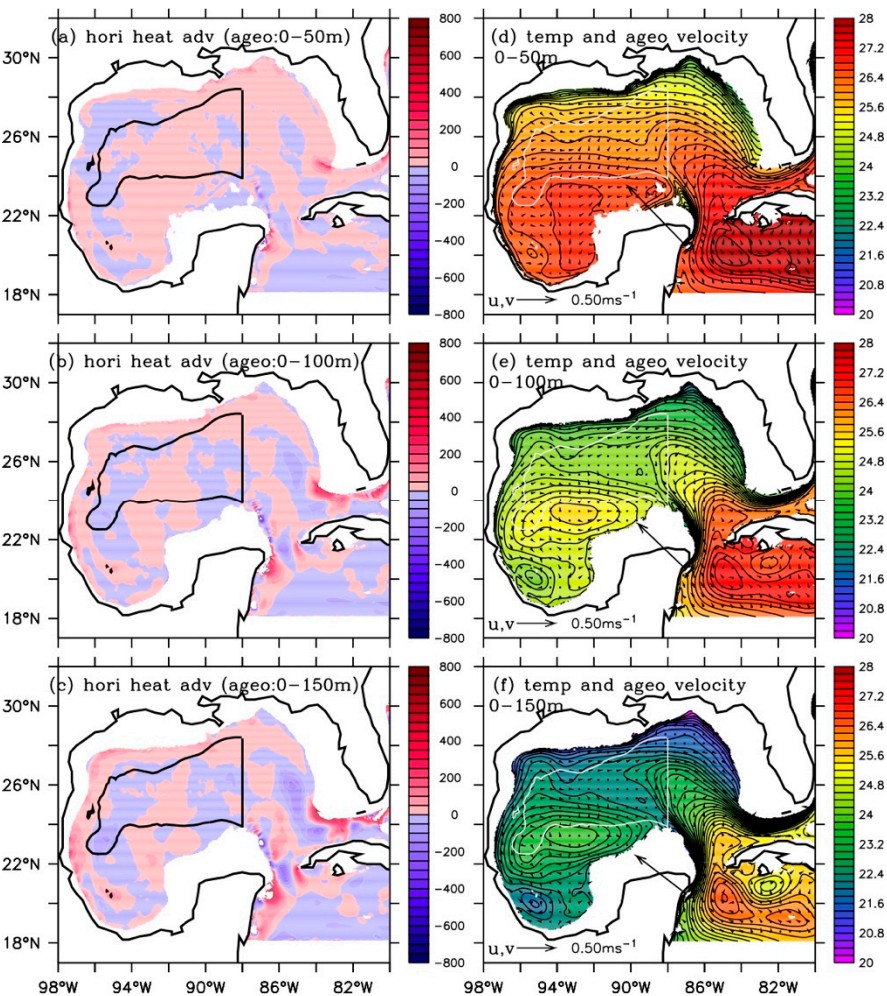

**Figure 6.** Mean horizontal heat advection caused by ageostrophic currents, arranged as in Figure 4.

The spatial pattern of the mean horizontal heat advection caused by the geostrophic current in each layer is similar to that in Figure 4, suggesting that the cooling in the open ocean of the western GOM (and in most of the eddy-active region) and the warming near the coastal shelf waters by the horizontal current arise primarily from the geostrophic current. By contrast, the horizontal heat advection by the ageostrophic current appears different from the geostrophic heat advection, showing a weak warming in the eddy-active region (particularly in the upper 50 m) and a warming in the western and northern shelf waters. The seasonally mean ageostrophic current is generally weaker than the seasonally mean geostrophic current, partly explaining why the geostrophic heat advection is overall greater than the ageostrophic heat advection at the seasonal timescale. The geostrophic and ageostrophic flows across the isotherms in the open ocean (e.g., in the eddy-active region) are generally weak, leading to an overall weak heat redistribution in the open ocean of the western Gulf by the horizontal seasonal mean flow. The weakness of the ageostrophic currents could be related to the smoothing of the wind forcing and the lack of currents induced by the Stokes drift. These factors are expected to make minor contributions to the layer-averaged horizontal thermal transport. Increased vertical motions in a two-way coupled model [54] are expected to enhance the vertical transport and are likely to have more impact on the horizontal transport than higher resolution winds or the Stokes drift.

### 5.3. Vertical Heat Advection

In this section, the contribution of the vertical heat advection to the heat redistribution in the three overlapping layers is examined (Figure 7). The positive values of the vertical

heat advection denote the downwelling warming is dominated relative to the upwelling cooling within the layers, and the negative values denote that the upwelling cooling is dominated relative to the downwelling warming within the layers. The vertical heat advection appears distinct from the horizontal heat advection in space distribution and magnitude. The magnitude of the vertical heat advection is much stronger than the horizontal heat advection, particularly for the 0–100 m and 0–150 m layers. In the eddy-active region (enclosed by the black curve in Figure 7), the downwelling warming (red in Figure 7) in the upper ocean is dominant. The strongest downward heat advection appears to be constrained to the trough of the western GOM. Upwelling cooling (blue in Figure 7) is mostly constrained to areas near the shelf and the shelf slopes. Figure 7d shows the two-dimensional mean vertical circulation (downward and upward motion) along the 25°N latitudinal circle, which partly explains the spatial pattern of downwelling warming and upwelling cooling. We hypothesize that this phenomenon is linked to the dominant anticyclonic wind stress curl [55] and/or the influence of the LCEs. Anticyclonic wind stress curl and/or the LCEs cause the downward motion in the primary region of the LCEs, leading to the upward motion in both edges of the trough in the central region of the western GOM due to orographic forcing. Coastal upwelling and downwelling may also play an important role in modulating the near-surface temperature near the continental shelf. The alternate spatial pattern of upward and downward motion leads to the alternate alignment of upwelling cooling and downwelling warming. The upwelling over the southwestern Gulf and the regions adjacent to the GOM's western boundary is also associated with the cyclonic wind stress curl over there [55].

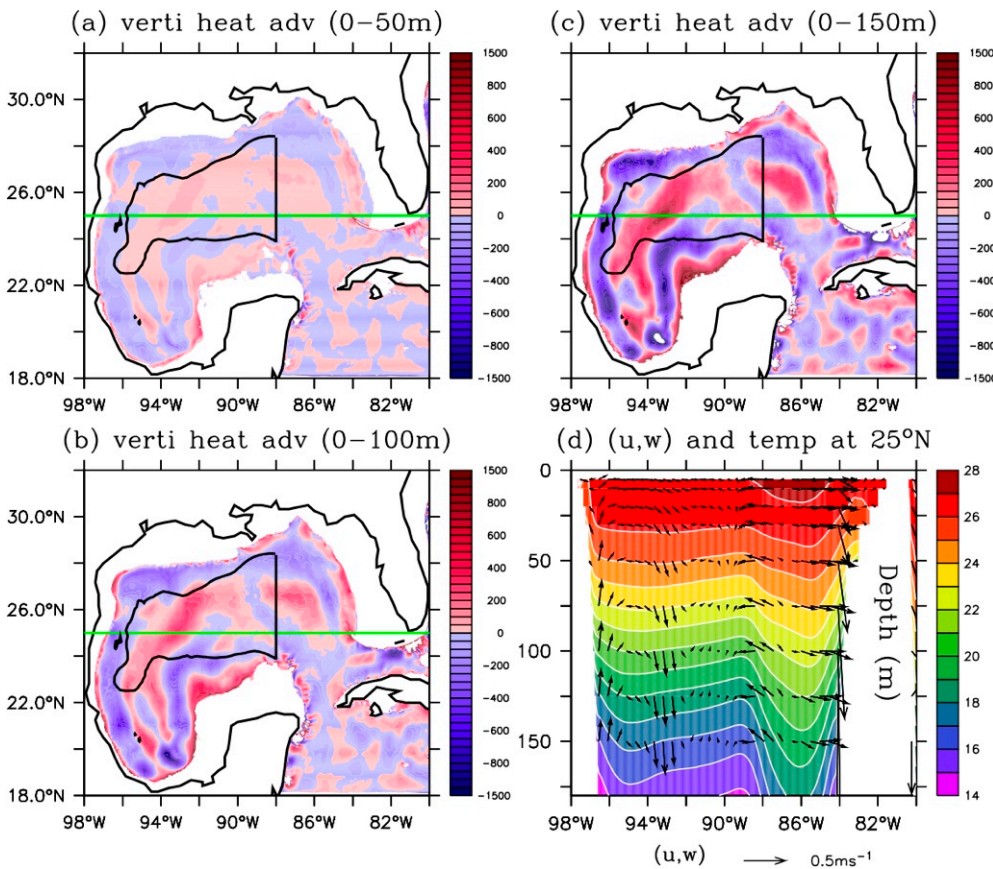

**Figure 7.** Mean vertical heat advection (W m$^{-2}$) in the upper (**a**) 50 m, (**b**) 100 m, and (**c**) 150 m. (**d**) Mean vertical circulation (m s$^{-1}$) and temperature (°C) along 25° N on x-z plane. Color scale in (**d**) is for temperature. Vertical velocity values in (**d**) have been multiplied by $10^4$, which is the ratio of longitudinal distance (98° W to 80° W) at 25° N to a depth of 180 m.

### 5.4. Eddy Heat Flux Convergence

The role of eddies in redistributing heat in three overlapping upper layers is discussed in this section. The majority of the eddy contribution is through the exchange of the eddy core water with the background rather than the eddy stirring the background. There is no doubt that eddies, including LCEs and cyclones, have a strong local impact on SST and/or near-surface temperature; however, the long-term mean impact over a large region has not been determined. This section seeks to address whether LCEs and cyclones have a long-term influence on the near-surface temperature in the western Gulf, particularly in the eddy-active region.

The convergence of eddy heat flux on the horizontal plane is calculated following the equation $-\left(\partial \overline{u'T'}/\partial x + \partial \overline{v'T'}/\partial y\right)$, where u′, v′, and T′ are deviations of the 5-day averaged total velocity and temperature from their seasonally averaged values. The overbar denotes the seasonal average as described in Section 3. Figure 8 shows the spatial distribution of the time-mean horizontal eddy heat flux convergence in the 0–50 m, 0–100 m, and 0–150 m layers from HYCOM. The positive and negative values (i.e., local warming of the upper layers) appear to be noisy in the western Gulf. The positive and negative values of the eddy heat flux convergence in the eddy-active region (demarcated by the solid black outline in Figure 8) represent the long-term impact of warm and cold eddies on the near-surface temperature in this region. Overall, the magnitudes of both the positive and negative values of the temporal mean eddy heat flux convergence in HYCOM are comparable to the mean horizontal heat advection by the seasonally mean flow.

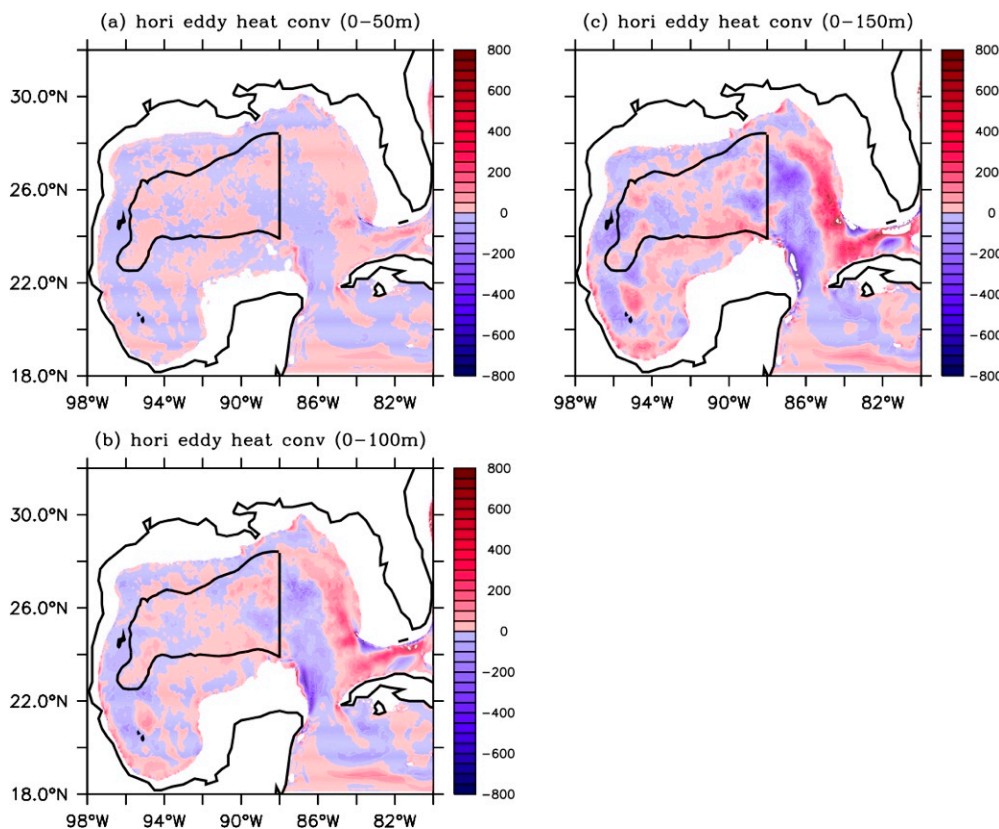

**Figure 8.** Maps of the time-mean horizontal eddy heat flux convergence (W m$^{-2}$) in the upper (**a**) 50 m, (**b**) 100 m, and (**c**) 150 m. Westward-migrating active eddies in the western GOM are primarily constrained in the surrounding region by the bold black curve, based on mean EKE spatial distribution derived from surface geostrophic velocity in AVISO. Positive values denote warming the ocean.

We also computed the vertical integral of the vertical eddy heat flux convergence $-\left(\partial \overline{w'T'}/\partial z\right)$ from depth $z_0$ to surface, which is determined by the turbulent vertical flux at the depth $z_0$ and at the surface (i.e., $(w'T')z_0 - (w'T')_0$) and show a map distribution of the long-term mean values for the three layers (Figure 9). Results indicate the long-term mean values are comparable to its horizontal component in magnitude and the negative and positive values are also noisy in space to some degree. As found in previous observational studies [22,56], the near-surface temperature can be effectively modulated locally by warm and cold eddies when they pass by, but the long-term impact of the eddy heat flux convergence on the near-surface temperature is not much larger than the horizontal advection by the mean current. Most importantly, the long-term mean eddy heat flux convergence is spatially noisy in some degree. This noisiness in space occurs partly because of the irregular life cycles and irregular paths of the warm and cold eddies. For example, although the LC eddies are detached from the LC from three to 17 months [57] and migrate westward, the westward migration of the LCEs can cause a short-term anomaly of the near-surface layer temperature. However, once the LC eddies propagate away, the convergence of the eddy heat flux becomes small locally and the local anomaly is mitigated by other processes, especially after the eddies weaken and dissipate. Statistically speaking, although the LC eddies can substantially affect the local near-surface temperature over a short period (e.g., within a life span of an individual LCE), the significant impact is greatly reduced when averaged over a long period and averaged over a large region (e.g., an eddy-active region) because of its irregular paths (the impact is reduced when averaged in space) and due to the irregular life cycles (the impact is reduced when averaged in time). As discussed in Section 5.5, the temporal mean of the eddy heat flux convergence averaged over the eddy-active region is small relative to its short-term variability.

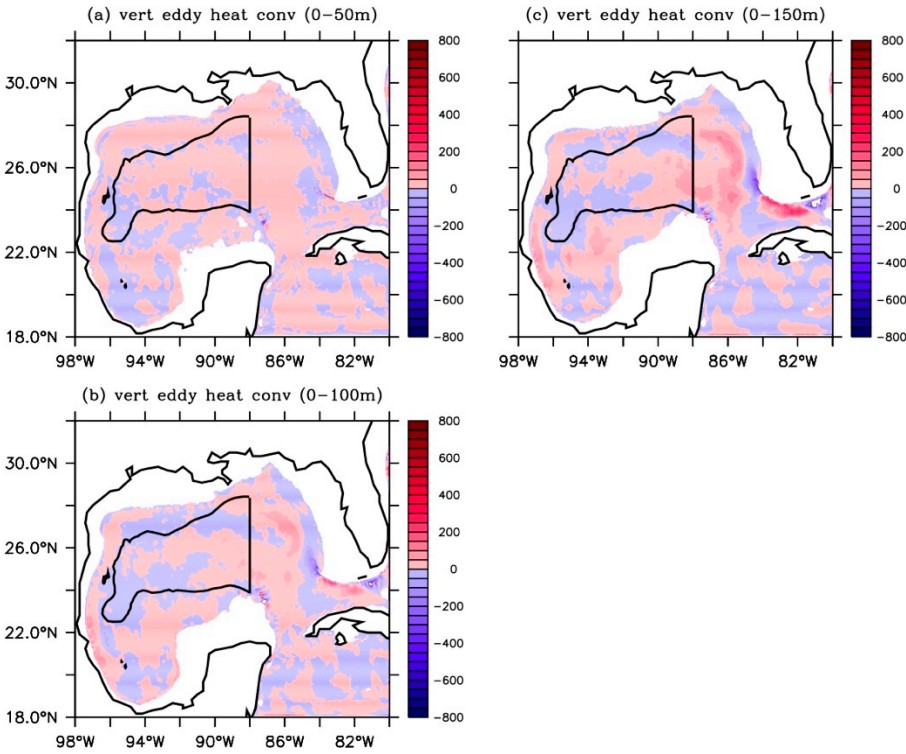

**Figure 9.** Same as Figure 8, except for mean vertical eddy heat flux convergence.

## 5.5. Subannual Variability of Heat Terms in the Eddy-Active Region and in the Western GOM

In Sections 5.1–5.4 we described the spatial variability of heat terms and found that not all of the heat terms have a coherent influence in space on the near-surface temperature. In this section, we will examine the temporal variability of each heat term and identify the

relative roles of the upper ocean physical processes in distributing heat in the upper ocean of the western GOM, particularly in the eddy-active region (an area indicated by the black solid curve in Figure 2). For this purpose, we compare the time series of the contributing terms averaged over a large region. Figure 10 illustrates the time series of the contributing terms averaged over the eddy-active region for the upper 50 m.

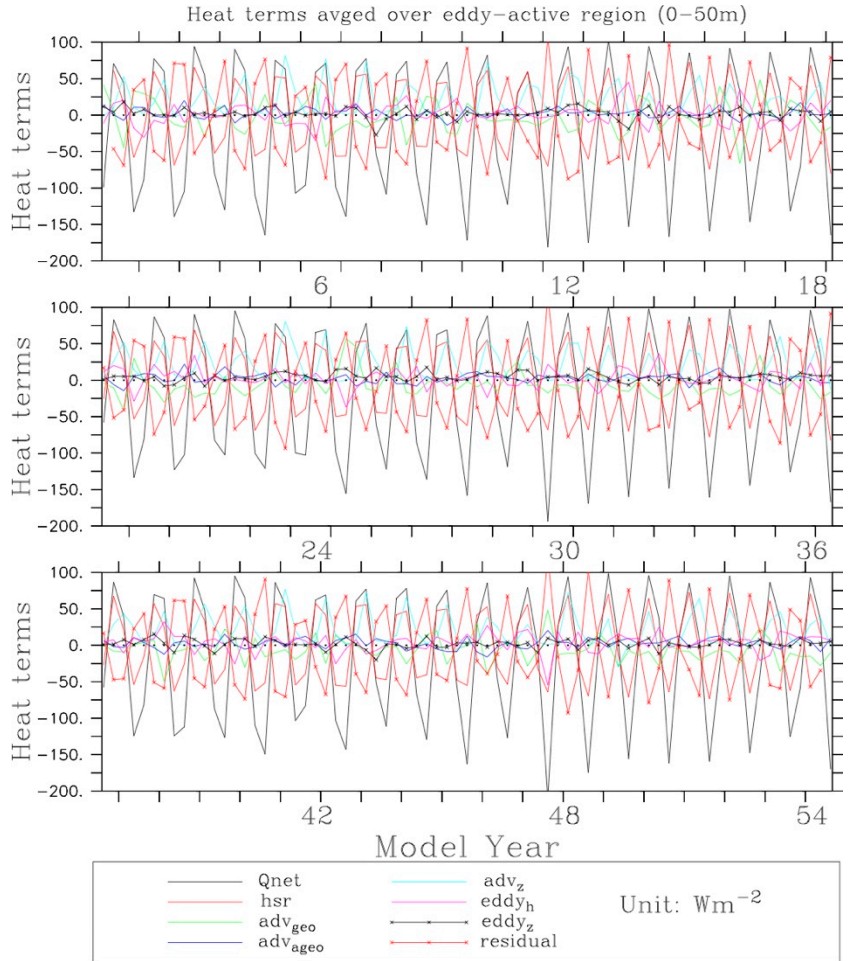

**Figure 10.** Heat budget analysis for the upper 50 m ocean. Time series of the net surface heat flux (Qnet, in black), rate of heat storage (hsr, in red), horizontal heat advection caused by geostrophic current (adv$_{geo}$, in green), horizontal heat advection caused by ageostrophic current (adv$_{ageo}$, in blue), vertical heat advection (adv$_z$, in cyan), horizontal eddy heat flux convergence (eddy$_h$, in magenta), vertical eddy heat flux convergence (eddy$_z$, in black-cross), and a residual term (in red-cross) area-averaged over an eddy-rich region. Unit: W m$^{-2}$.

In the eddy-active region, the seasonal cycle of both the net surface heat flux (in black) and the rate of heat storage (in red) is robust. Evolution of the heat storage rate appears to almost to synchronize with the net surface heat flux, with a maximum warming in summer and a minimum cooling in winter. The negative value of temporal mean net surface heat flux over the eddy-active region is because the heat loss from the ocean into the atmosphere during winter and spring seasons is greater than the heat gain from the atmosphere during summer and fall seasons. Vertical heat advection by the mean flow (in cyan) also has a robust seasonal cycle and contributes most to the warming in the upper layer, with a maximal warming in winter and spring and a minimal warming in summer. The strong seasonal cycle of vertical heat advection is primarily induced by the strong seasonal cycle of mixed layers. In summer, the net surface heat flux is large, and the winds are weak, leading to a shallow mixed layer and a shallow layer of strong stratification that tends

to prohibit downwelling and vertical mixing. This causes relatively weak vertical heat advection. In winter, the strong negative surface heat flux and strong winds cause a deeper mixed layer, leading to strong downwelling or upwelling. This, in turn, leads to relatively strong vertical heat advection. This will be further discussed in Section 5.6. The downward heat transport in the upper ocean of eddy-active region is presumably caused by the overall sinking (i.e., downward motion) in the western GOM due to the overall anticyclonic wind stress curl [55,58]. This sinking in the upper layer of the western Gulf was proposed in study [59], which found a net upper-layer mean flow to the west in the central GOM from observations, which requires that there be sinking in the upper-layer of the western Gulf. The temporal mean rate of heat storage is expected to be small.

The temporal variability of the area-averaged horizontal heat advection caused by the geostrophic current (in green) is comparable to the vertical heat advection but has no robust seasonal cycle. The geostrophic current tends to contribute to an overall cooling in the upper 50 m ocean in the eddy-active region. Horizontal advection by the ageostrophic current (in blue), as well as the horizontal (in magenta) and vertical (in black-cross) eddy heat flux convergence, is weaker relative to the strong seasonal variability of the heat storage rate (in red), the net surface heat flux (in black), and the vertical heat advection (in cyan). Both the horizontal heat advection and the eddy heat flux convergence do not have a robust seasonal cycle. The evolution of residual heat (in red-cross) tends to be balanced primarily by the net surface heat flux, the heat storage rate, and the vertical heat advection. Furthermore, both the area-averaged eddy heat flux convergence and the horizontal heat advection have no clear relationship with the net surface heat flux, suggesting that the eddy activity and the horizontal mean flow affect the heat distribution in the upper 50 m layer in an irregular way (i.e., the phase is not locked by season and the peak values can happen in any seasons), which is different from the way that the net surface heat flux and the vertical heat advection influence the heat distribution, because both the net surface heat flux and the vertical heat advection have a robust seasonal cycle (i.e., the phase is locked). The phase-lock relation is established in a way that the net surface heat flux overall causes a strong warming to the upper ocean during summer, as the surface heat from the net surface heat flux is transported downward through a weak vertical heat advection, inhibiting the downward heat transfer to deep layers; thus, the heat from the net surface heat flux is stored in the upper ocean during these two seasons. During winter, the negative surface heat flux implies that the heat stored in the upper ocean during summer is released to the atmosphere. The resultant cooling in the upper ocean from the negative net surface heat flux and the strong winds tend to destroy the stratification in the upper ocean, favoring downwelling processes, ultimately leading to relatively strong vertical heat advection during winter. It should be noted that the lasting positive value of the eddy heat flux convergence (in magenta) can be shorter or longer than one year, which is consistent with the observed lifespan of LCEs. When these contributing terms are averaged over several years, we expect an overall warming in the upper 50 m layer caused by upper-ocean physical processes (i.e., the mean heat advection by the mean horizontal and vertical flow, and the eddy heat flux convergence by the eddies) to be lost to the atmosphere through the negative heat flux. The exchange between the top 50 m and the rest of the mixed layer is important in winter, when the mixed layer is deep and it can reach 150 m [60], while during summer, when the mixed layer is shallow and the seasonal thermocline is sharp, the vertical exchanges between the top 50 m and the water below may be limited. However, the above results show that the overall variability of the area-averaged heat redistribution by the horizontal heat advection and the eddy heat flux convergence in the upper 50 m of the eddy-active region is smaller than the net surface heat flux and the vertical heat advection. The small magnitude of the area-averaged horizontal heat advection can be attributed to the fact that the temporal mean flow slightly crosses the temporal mean isotherm (Figure 4d) and the temporal seasonally mean flow is primarily from the seasonally mean geostrophic current (i.e., the seasonally mean ageostrophic current is relatively weak), as discussed in Section 5.2. The small magnitude of the area-averaged eddy heat flux convergence is

attributed to the noisiness in space and the irregular life spans of LCEs and cyclones, as discussed in Section 5.4. It may also be due to the fact that the local warming at one location caused by the passing of warm rings in one period can be offset by the local cooling that occurs when cold rings pass by during another period. Figure 11 is the same as Figure 10, except for 0–150 m, and the above results are similar for 0–150 m, except that the seasonal cycle of the vertical heat advection becomes relatively weak. The reduced seasonal cycle of the vertical advection of heat is expected because the deeper ocean is less affected by the seasonal variability of the surface heat fluxes and the overlying atmospheric winds. The time-mean and the standard deviation of the contributing terms for the 0–150 m layer in the eddy-active region are summarized in Table 2. The cooling from the mean horizontal heat advection in the eddy-active region in the upper 150 m becomes greater and is comparable to net surface heat flux, in which the geostrophic current is more important than the ageostrophic current in cooling the ocean in the eddy-active region (Table 2). The diffusive heat flux is not small because it occurs across the thermocline, where the strong vertical temperature gradient is present [61]. However, the diffusive heat flux is assumed to mainly influence the temperature of the mixed-layer bottom, not the near-surface temperature or SST.

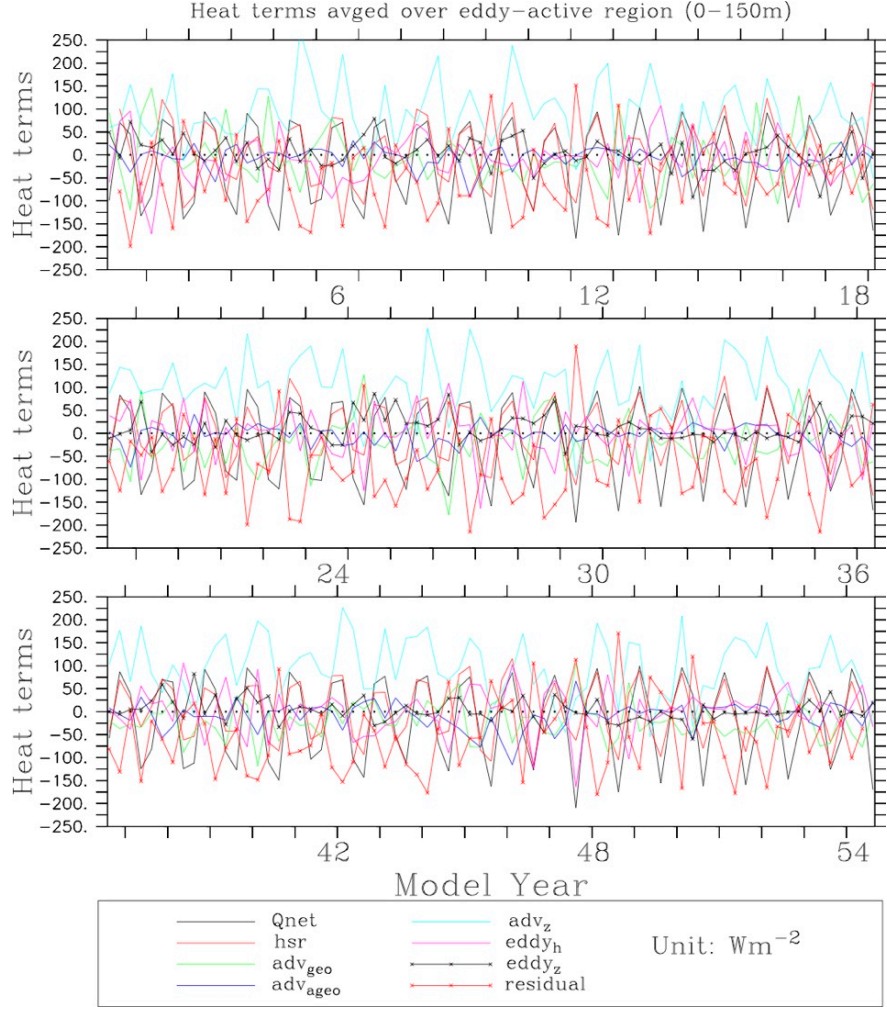

**Figure 11.** Same as Figure 10 except for the upper 150 m ocean.

**Table 2.** Mean $\pm$ std of ocean heat budget terms (hsr = Qnet + $adv_h$ + $adv_z$ + $eddy_h$ + $eddy_z$ + residual in Equation (2)) for the 0–150 m layers averaged over an eddy-active region (EDDY in short) where the observed surface eddy kinetic energy is larger than 200 cm$^2$ s$^{-2}$ based on surface AVISO data, which is illustrated in a region enclosed by bold black curves in Figure 2, and over the western GOM (WGOM in short) (98°W–88°W,18°N–30°N). Only the heat terms in the region where waters are shallower than 150 m are computed. Note that $adv_h$ = $adv_{geo}$ + $adv_{ageo}$. (Units: W m$^{-2}$) Abbreviations: hsr: heat storage rate; Qnet: net surface heat flux; $adv_h$: horizontal heat advection by seasonal mean current; $adv_z$: vertical heat advection by seasonal mean current; $adv_{geo}$: horizontal heat advection by seasonal mean geostrophic current; $adv_{ageo}$: horizontal heat advection by seasonal mean ageostrophic current; $eddy_h$: horizontal eddy heat flux convergence; $eddy_z$: vertical eddy heat flux convergence.

| Regions | hsr | Qnet | $adv_h$ | $adv_{geo}$ | $adv_{ageo}$ | $adv_z$ | $eddy_h$ | $eddy_z$ | Residual |
|---|---|---|---|---|---|---|---|---|---|
| EDDY | 0.4 $\pm$ 66 | $-18.9 \pm 91$ | $-23.3 \pm 61$ | $-16.9 \pm 51$ | $-6.4 \pm 25$ | 95.4 $\pm$ 108 | $-3.9 \pm 51$ | 5.4 $\pm$ 27 | $-54.3 \pm 120$ |
| WGOM | 0.5 $\pm$ 58 | $-10.8 \pm 89$ | 12.6 $\pm$ 20 | 5.9 $\pm$ 16 | 6.7 $\pm$ 11 | $-7.1 \pm 29$ | $-0.3 \pm 12$ | 5.5 $\pm$ 13 | 0.6 $\pm$ 46 |

We also examine the temporal variability of the heat terms in the upper 50 m averaged over the whole western Gulf (west of 88° W; Figure 12). The heat terms in the regions where the waters are not deeper than 50 m are excluded in Figure 12 because it is shallower than the integration depth. It is not surprising to see the robust seasonal variability of the heat storage rate and the net surface heat flux, as found in the eddy-active region. However, the temporal variability of the horizontal and vertical heat advection and the eddy heat flux convergence within the top 50 m layer are much weaker when averaged over the whole western Gulf compared to those averaged over the eddy-active region. Vertical heat advection becomes much smaller relative to that in the eddy-active region, and the magnitude is of the same order of the horizontal heat advection. The dramatic decrease in the temporal variability of the horizontal and vertical heat advection and the eddy heat flux convergence suggests that the influence of the ocean physical processes on the near-surface temperature in the eddy-active region can be largely compensated/offset by the ocean processes in the remaining region. Similar to the heat terms in the eddy-active region, the seasonal variability of the heat terms in the western GOM is also reduced when integrated over the 0–150 m layer (Table 2).

Table 2 also illustrates the time-mean and standard deviation values of the heat terms for the 0–150 m layer when averaged over the western GOM. In contrast to the eddy-active region, the time-mean vertical heat advection term, when averaged over the entire western Gulf, is much smaller and becomes negative (i.e., contributing to an overall cooling for the entire western Gulf) due to the dominant upward heat flux outside the eddy-active region. Other heat terms are changed as well. For example, instead of causing cooling in the eddy-active region, the mean horizontal heat advection contributes to warming when averaged over the western Gulf, which is almost equally contributed to from the geostrophic and ageostrophic currents. This implies that while the geostrophic current in the eddy-active region is dominated in the horizontal heat advection, the ageostrophic current is dominated in the horizontal heat advection in the remaining region. These results suggest that the role of the upper-ocean processes in affecting the near-surface temperature in the eddy-active region is different from those over the remaining regions, probably associated with the dominant anticyclonic wind stress curl (negative values) over the eddy-active region and the dominated cyclonic wind stress curl (positive values) over the southwestern portions of western Gulf [55]. The sum of the horizontal and the vertical eddy heat flux convergence is on the same order of the horizontal heat advection. Warming from the horizontal advective heat flux over the upper 150 m is dominated in an area other than the eddy-active region, and the warming is larger than the cooling from the horizontal advective heat flux in the eddy-active region, which is consistent with the results in Figure 5. Table 2 indicates a small positive value (+ 0.6 Wm$^{-2}$) of the residual term when averaged over the western GOM. This happens because the positive value of the solar radiation across the 150 m depth may

slightly exceed the negative value of the heat diffusion. Note that, except for heat storage rate and net surface heat flux, the variability of the heat terms in the eddy-active region is generally greater than that when averaged over the entire western GOM (Table 2).

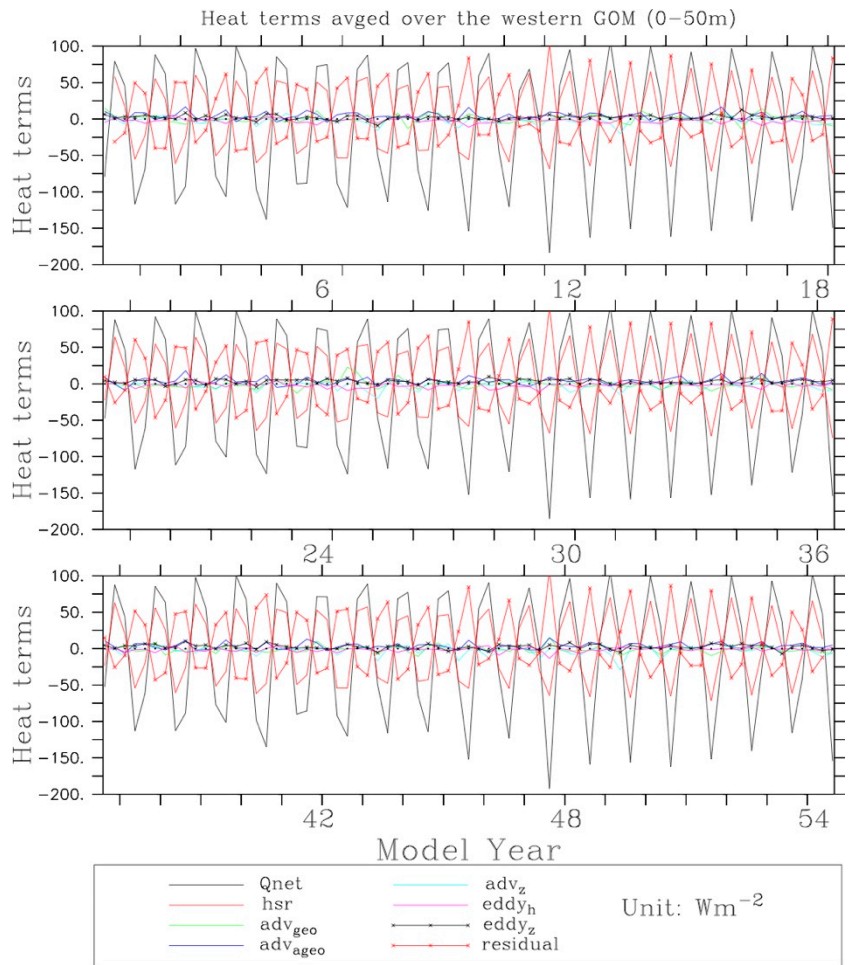

**Figure 12.** Same as Figure 10 except for heat terms area-averaged over the whole western GOM.

In summary, the above results suggest that the effects of the upper-ocean processes on the near-surface temperature/heat content over the eddy-active region (deep ocean) are different from those over the remaining region of the western GOM (shallow ocean).

### 5.6. Heat Budget in Summer and Winter Seasons

As discussed in the previous sections, the upper-ocean processes not only play a different role in the heat redistribution over the eddy-active region and over the remaining region of the western GOM, but also play a different role in heat redistribution in summer and winter. Some of the heat terms appear to have a robust seasonal cycle, partly associated with the strong seasonal cycle of mixed layer processes, as discussed in Section 5.5. To further understand the distinct roles of the upper-ocean physical processes during summers and winters, the heat budget terms are computed in summers (June–July–August; JJA) and in winters (December–January–February; DJF), whose map distributions are illustrated in Figures 13 and 14, respectively.

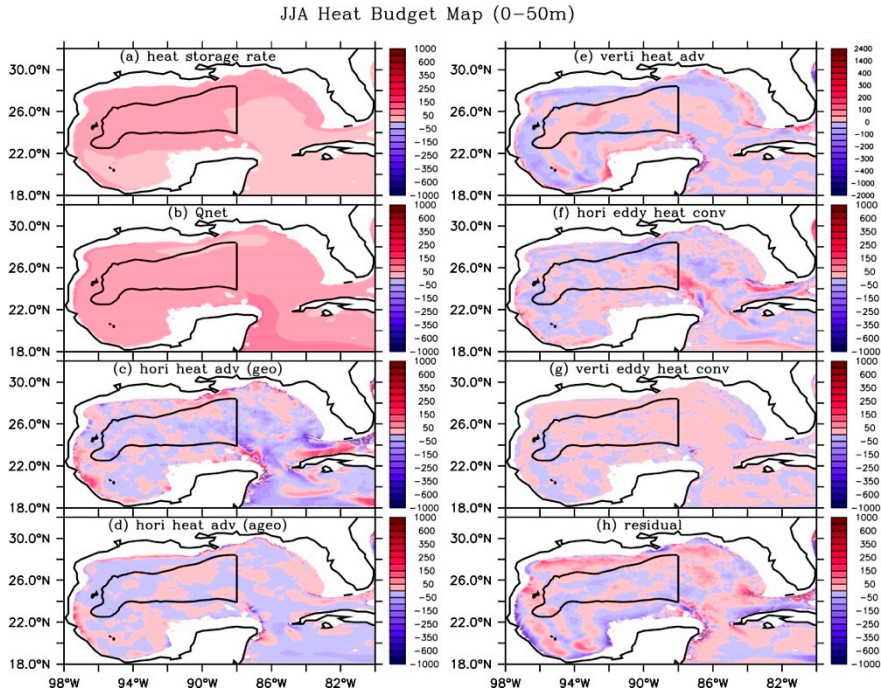

**Figure 13.** Mean heat budget terms (**a**) heat storage rate, (**b**) net surface heat flux Qnet, (**c**) horizontal heat advection by geostrophic current, (**d**) horizontal heat advection by ageostrophic current, (**e**) vertical heat advection, (**f**) horizontal eddy heat flux convergence, (**g**) vertical eddy heat flux convergence, and (**h**) the residual term in the top 50 m time-averaged over summer season (June–July–August; JJA). Only region deeper than 50 m is plotted. Unit: W m$^{-2}$.

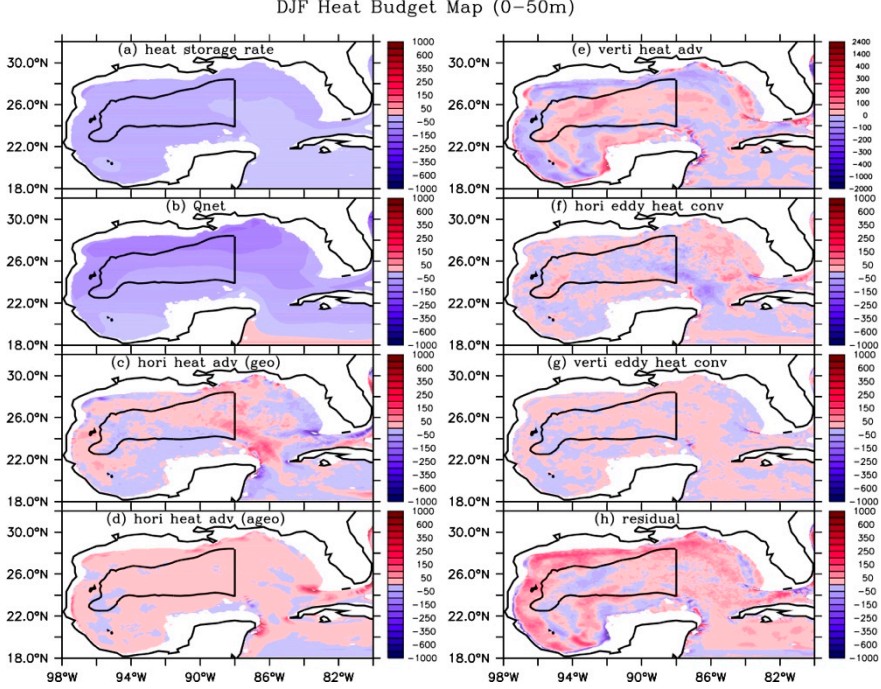

**Figure 14.** Same as Figure 13 but for those time-averaged over winter season (December–January–February; DJF).

The map distributions of the heat terms for the top 50 m in summers and in winters appear quite different, particularly for those with a robust seasonal cycle, such as the net surface heat flux, heat storage rate, and vertical heat advection, as discussed in

Section 5.5. Both the net surface heat flux and the heat storage rate warm the Gulf in summer (Figure 13a,b) and cool the Gulf in winter (Figure 14a,b). Both the downward advective heat flux in the eddy-active region and the upward advective heat flux outside of the eddy-active region (Figure 14e) are stronger in winters than their counterparts in summers (Figure 13e), partly due to the differences in the mixed layer processes between summers and winters. The seasonal variability of the mixed layer depth is largely induced by the differences in the net surface heat flux and the overlying surface winds between summers and winters. Horizontal heat advection, owing to geostrophic and ageostrophic currents overall, cause cooling in summers (Figure 13c,d) and warming in winters (Figure 14c,d) in the eddy-active region. Both the horizontal and vertical eddy heat flux convergences in summers are weak compared to the net surface heat flux, heat storage rate, and vertical heat advection in the same season. Note that the horizontal eddy heat flux convergence appears noisy in space in both summers and winters. The above features regarding the heat redistribution induced by the upper-ocean physical processes hold for the upper 100 m and 150 m layers (not shown) except that the values get larger as the heat terms are integrated into the deeper layers. Because the residual heat includes both the positive values of the downward solar radiation penetrating across the 50 m depth and the negative values of the heat diffusion at the 50 m depth, the positive (negative) values of the residual heat at one location, shown in Figures 13h and 14h, suggests that the surplus of solar radiation (heat diffusion) exceeds the heat diffusion (the surplus of solar radiation) at that location.

Table 3 is the same as Table 2 except for the heat budget terms in the upper 150 m spatially-averaged in the eddy-active region and the western GOM and the temporally averaged over the JJA and DJF periods, respectively. It is clear that the upper-ocean processes play a distinctive role between summers and winters over both the eddy-active region and over the whole western GOM. For example, the vertical heat advections in both regions tend to be slightly stronger during winters than during summers. Horizontal heat advection causes strong cooling in summers and weak warming in winters over the eddy-active region. When averaged over the entire western GOM, the horizontal heat advection contributes to warming the ocean in both summers and winters. The distinct features in the horizontal heat advection between the eddy-active region (deep ocean) and the remaining region of the western GOM (shallow ocean) in the same season are attributed to the different roles played by the mean horizontal flow (including the geostrophic and ageostrophic current) and the vertical velocity in the above two regions, as listed in Table 3. The interannual variability of the heat terms is greater than those averaged over the entire western GOM for both summer and winter seasons (Table 3).

**Table 3.** Same as Table 2 except averaged over summer (JJA) and winter (DJF) seasons.

| Regions | hsr | Qnet | $adv_h$ | $adv_{geo}$ | $adv_{ageo}$ | $adv_z$ | $eddy_h$ | $eddy_z$ | Residual |
|---------|-----|------|---------|-------------|--------------|---------|----------|----------|----------|
| JJA | | | | | | | | | |
| EDDY | $80.6 \pm 39$ | $65.5 \pm 7$ | $-50.6 \pm 66$ | $-33.2 \pm 47$ | $-17.4 \pm 30$ | $90.2 \pm 45$ | $-2.1 \pm 61$ | $6.8 \pm 20$ | $-29.2 \pm 56$ |
| WGOM | $74.7 \pm 19$ | $68.1 \pm 6$ | $14.7 \pm 20$ | $6.5 \pm 14$ | $8.2 \pm 12$ | $-2.0 \pm 14$ | $-2.9 \pm 15$ | $5.8 \pm 9$ | $-9.0 \pm 26$ |
| DJF | | | | | | | | | |
| EDDY | $-91.3 \pm 36$ | $-115.0 \pm 25$ | $3.5 \pm 58$ | $-1.1 \pm 54$ | $4.6 \pm 15$ | $105.5 \pm 51$ | $-15.6 \pm 54$ | $5.6 \pm 22$ | $-75.3 \pm 62$ |
| WGOM | $-94.5 \pm 21$ | $-103.3 \pm 22$ | $19.3 \pm 17$ | $9.4 \pm 15$ | $9.9 \pm 7$ | $-11.6 \pm 16$ | $-1.9 \pm 10$ | $6.6 \pm 12$ | $-3.6 \pm 25$ |

## 6. Summary and Conclusions

This study investigates the relative roles of the upper-ocean physical processes in distributing heat in the near-surface layer in the western GOM using a 54-year simulation of the eddy-resolving HYCOM and observational datasets. The major terms in the Reynolds averaging heat equation for 0–50 m, 0–100 m, and 0–150 m layers of the western GOM are computed and compared in sign and magnitude. Two caveats are worth pointing out. First,

the vertical velocity is a diagnostic variable derived from the HYCOM horizontal velocity. Second, to estimate the heat terms for the constant layer of an upper ocean, the horizontal velocity and temperature were interpolated to Cartesian coordinates from the hybrid coordinate system of the original HYCOM grids using HYCOM's postprocessing package.

With these caveats in mind, three major results are presented in this study:

(1) The mean heat budget analysis in 0–150 m layer in the eddy-active region indicates that the net surface heat flux, the horizontal heat advection, and the vertical heat advection make the dominant contributions to the long-term heat budget of the upper layer. Cooling from diffusive processes mainly affect the water temperature between the mixed-layer bottom and thermocline, not the near-surface temperature or SST. Vertical heat advection has a robust seasonal variability, which is supposed to be associated with the strong seasonal cycle of the mixed layer and the strong seasonal cycle of the net surface heat flux which provides persistent warming (cooling) to the surface ocean in summer (winter). Horizontal heat advection contributes to overall cooling ($\sim -23$ W m$^{-2}$), which is primarily caused by the geostrophic current ($\sim -17$ W m$^{-2}$). The eddy heat flux convergence contributes to the overall warming in the eddy-active region. The time-mean area-averaged eddy heat flux convergence is small (–1.5 W m$^{-2}$) relative to the short-term variability (at an order of 10 W m$^{-2}$, see Table 2 and Figure 11).

(2) The mean heat budget analysis in the whole western GOM is compared to those in the eddy-active region. Analysis suggests that the upper-ocean physical processes play a distinct role in the near-surface temperature change between the eddy-rich region and the remaining region of the western GOM. When averaging in space, the warming from the downward heat advection in the eddy-active region due to the dominant anticyclonic wind stress curl and/or the LCE downwelling over the eddy-active region can be largely canceled by the cooling from the upward advective heat in the remaining region due to the cyclonic (positive) wind stress curl over the southwestern part of the western Gulf (Gutierrez de Velasco and Winant, 1996) and/or the orographic effects near the shelf slope of the Gulf shelf (Figure 7). The variability of the heat terms in the eddy-active region is larger than that when averaged over the entire western GOM (Table 2).

(3) Heat budget analyses for summer and winter seasons reveal that the net surface heat flux, the heat storage rate, and the vertical heat advection have distinctive features in redistributing heat due to their strong seasonal cycles. Net surface heat flux and heat storage contribute to an overall warming in the entire Gulf during summers and an overall cooling during winters. Horizontal heat advection contributes to an overall warming in the entire GOM in both summers and winters. However, the horizontal heat advection contributes to an overall cooling in summers and to an overall weak warming in the eddy-active region during winters. Downward heat advection (warming the ocean) is dominant in the eddy-active region, with the maximum in winters and the minimum in summers, presumably associated with the robust seasonal cycle of the mixed-layer depth. Upward heat advection (cooling the ocean) is dominant in the remaining regions of the western GOM, with the maximum in winters and the minimum in summers. The eddy heat flux convergence contributes to an overall warming in the whole western GOM in summers and winters. When averaged over the eddy-active region, eddies tend to cool (warm) the upper ocean in winters (summers). The interannual variability of the heat terms in the eddy-active region is greater than that when averaged over the entire western GOM for both summer and winter seasons (Table 3).

**Author Contributions:** Conceptualization, Y.Z.; methodology, Y.Z., D.D. and M.A.B.; validation, Y.Z., D.D. and M.A.B.; formal analysis, Y.Z.; data curation, Y.Z. and D.D.; writing—original draft preparation, Y.Z.; writing—review and editing, D.D., M.A.B. and M.M.A. All authors have read and agreed to the published version of the manuscript.

**Funding:** This research was funded by the U.S. National Aeronautics and Space Administration (NASA) Physical Oceanography of the Ocean Vector Winds Science Team (OVWST) and the U.S.

National Oceanic and Atmospheric Administration (NOAA) Global Ocean Monitoring and Observing (GOMO).

**Institutional Review Board Statement:** Not Applicable.

**Informed Consent Statement:** Not Applicable.

**Data Availability Statement:** All datasets used in this study can be obtained from the public sources mentioned in the acknowledgements.

**Acknowledgments:** We are greatly indebted to all those who contributed to the observations and global datasets used in this study. The HYCOM model output fields are available at https://www.hycom.org/data/goml0pt04/expt-02pt2, accessed on 17 March 2022, which was provided by Dmitry Dukhovskoy. AVHRR data were made freely available by NOAA and NASA and can be obtained through anonymous File Transfer Protocol (FTP) (ftp://ftp.nodc.noaa.gov/pub/data.nodc/pathfinder/Version5.2/, accessed on 17 March 2022). The altimeter products were produced by Ssalto/Duacs and distributed by AVISO, with support from CNES (http://www.aviso.altimetry.fr/duacs/, accessed on 17 March 2022). Real-time OSCAR ocean surface currents can be obtained at: https://podaac.jpl.nasa.gov/dataset/OSCAR_L4_OC_third-deg, accessed on 17 March 2022. GDEM3 ocean temperature monthly climatology are provided by the U.S. Navy (https://accession.nodc.noaa.gov/download/9600094, accessed on 17 March 2022). OAFlux data products are created by the Woods Hole Oceanographic Institution (WHOI) and are available through the FTP, available online: (ftp://ftp.whoi.edu/pub/science/oaflux/data_v3, accessed on 17 March 2022). ISCCP-FD radiative flux products are available and can be obtained at: https://isccp.giss.nasa.gov/projects/flux.html, accessed on 17 March 2022). Computational facilities have been provided by the Center for Ocean-Atmospheric Prediction Studies (COAPS) at Florida State University. Yangxing Zheng was supported by the U.S. National Aeronautics and Space Administration (NASA) Physical Oceanography of the Ocean Vector Winds Science Team (OVWST). Mark Bourassa was supported by the NASA OVWST and NOAA GOMO. All figures were made using the Ferret package version 6.3, which is provided by the Science Data Integration Group in the Pacific Marine Environmental Laboratory, NOAA.

**Conflicts of Interest:** The authors declare no conflict of interest.

## Appendix A

The contribution of the individual terms to the upper-ocean heat content variability is examined by using the energy equation

$$\rho C_p \left[ \frac{\partial T}{\partial t} + (V \cdot \nabla)T \right] = \rho C_p (\nabla \cdot K \nabla T) + Q_{sol} \gamma(z) + \varepsilon \tag{A1}$$

where T is ocean temperature, $C_p$ is the specific heat at constant pressure, V (u, v, w) is the ocean velocity vector, K is thermal diffusivity, and $\varepsilon$ is viscous dissipation, not considered here. $Q_{sol}$ is the net downward solar radiation that attenuates with depth, described by $\gamma(z)$. In the ocean, red and near-infrared radiation is absorbed in the top near-surface layer, whereas the visible and ultraviolet radiation penetrates below down to −100 m [62]. HYCOM allows several options for approximating the depth of penetration of the solar radiation ($\gamma(z)$). In the analyzed experiment, the Jerlov-like algorithm by Kara et al. [63] was employed. In this scheme, −98% of the red spectrum is absorbed within the top 2 m of the ocean.

Next, to investigate the contribution of the mesoscale eddies on the upper-ocean heat content, V and T are decomposed into the mean (time-averaged) and perturbation term

$$T = \overline{T} + T' \tag{A2}$$

$$V = \overline{V} + V' \tag{A3}$$

where the overbar denotes the mean, and the prime denotes the perturbations. The mean terms are derived by averaging the T and V fields over 90 days, as explained in Section 3.2. The perturbation terms are derived by deviating the V and T from the 90-day mean $\overline{V}$ and $\overline{T}$.

The mean and perturbation terms (A2) and (A3) are substituted into (A1). After applying the Reynolds averaging and vertically integrating over the upper $z_0$ m, the upper-ocean heat budget equation with the mean and perturbation components is derived

$$\rho C_p \int_{z_0}^{0} \frac{\partial \overline{T}}{\partial t} dz = -\rho C_p \left[ \int_{z_0}^{0} \left( \overline{u} \frac{\partial \overline{T}}{\partial x} + \overline{v} \frac{\partial \overline{T}}{\partial y} \right) dz + \int_{z_0}^{0} \left( \frac{\partial \langle \overline{u'T'} \rangle}{\partial x} + \frac{\partial \langle \overline{v'T'} \rangle}{\partial y} \right) dz \right.$$
$$+ \int_{z_0}^{0} \overline{w} \frac{\partial \overline{T}}{\partial z} dz + \int_{z_0}^{0} \frac{\partial \langle \overline{w'T'} \rangle}{\partial z} dz - \int_{z_0}^{0} \nabla_h \kappa_h \nabla_h \overline{T} dz \qquad \text{(A4)}$$
$$\left. - \int_{z_0}^{0} \frac{\partial}{\partial z} \kappa_v \frac{\partial \overline{T}}{\partial z} dz \right] + \int_{z_0}^{0} Q_{sol} \gamma(z) dz$$

where $\kappa_h$ and $\kappa_v$ are horizontal and vertical thermal diffusion coefficients. Equation (A4) describes the heat content change in the upper-ocean layer from z0 to the surface. The first two terms on the left-hand side describe the horizontal thermal fluxes due to the mean and perturbation components. The third and the fourth terms need further consideration to aid in analyzing the upper-ocean heat content change. The third term can be written as

$$\int_{z_0}^{0} \overline{w} \frac{\partial \overline{T}}{\partial z} dz = \int_{z_0}^{0} \left( \frac{\partial \overline{w}\overline{T}}{\partial z} - \overline{T} \frac{\partial \overline{w}}{\partial z} \right) dz \qquad \text{(A5)}$$

Using the kinematic boundary condition for $\overline{w}$ on the surface

$$\overline{w} = 0, \ z = 0 \qquad \text{(A6)}$$

(A5) becomes

$$\int_{z_0}^{0} \left( \frac{\partial \overline{w}\overline{T}}{\partial z} - \overline{T} \frac{\partial \overline{w}}{\partial z} \right) dz = -\overline{w}(z_0) \overline{T}(z_0) - \int_{z_0}^{0} \overline{T} \frac{\partial \overline{w}}{\partial z} dz \qquad \text{(A7)}$$

The last integral on the right-hand side of (A7) can be approximated following [6] as

$$\int_{z_0}^{0} \overline{T} \frac{\partial \overline{w}}{\partial z} dz \approx \int_{z_0}^{0} \hat{T} \frac{\partial \overline{w}}{\partial z} dz = -\hat{T}\overline{w}(z_0) \qquad \text{(A8a)}$$

where $\hat{T}$ is the depth-averaged temperature in the upper-ocean layer above $z_0$. Combining (A7) and (A8a), the vertically integrated (A5) is

$$\rho C_p \int_{z_0}^{0} \overline{w} \frac{\partial \overline{T}}{\partial z} dz = \rho C_p (\hat{T} - \overline{T}(z_0)) \overline{w}(z_0). \qquad \text{(A8b)}$$

The term describes heat flux across the bottom surface at $z_0$ due to the mean vertical velocity and the upper-ocean temperature (note the negative sign in front of this term in (A4)). The fourth term on the right-hand side can be written as

$$\rho C_p \int_{z_0}^{0} \frac{\partial \left( \overline{w'T'} \right)}{\partial z} dz = \rho C_p \left( \left( \overline{w'T'} \right)_0 - \left( \overline{w'T'} \right)_{z_0} \right) \qquad \text{(A8c)}$$

where $\eta$ is the sea surface height.

$$w' = \frac{\partial \eta}{\partial t}, \ \text{at } z = 0. \qquad \text{(A8d)}$$

Next, integration of the vertical diffusion term can be simplified as

$$\rho C_P \int_{z_0}^{0} \frac{\partial}{\partial z} \kappa_v \frac{\partial \overline{T}}{\partial z} dz = Q_{net} - \rho C_P \kappa_v \left. \frac{\partial \overline{T}}{\partial z} \right|_{z0} \tag{A9}$$

where $Q_{net}$ is the net surface heat flux at the surface (the sum of the downward net longwave radiation, and the downward turbulent latent and sensible heat fluxes). The last term on the right-hand side describes the thermal diffusion across the surface at $z_0$.

Combining all steps, the equation describing the upper-ocean heat content in terms of the mean and perturbation fluxes is obtained

$$\begin{aligned} \rho C_P \int_{z_0}^{0} \frac{\partial \overline{T}}{\partial t} dz = \quad & Q_{net} - \rho C_P \left[ \int_{z0}^{0} \left( \overline{V}_h \cdot \nabla_h \overline{T} + \nabla_h \cdot \left( \overline{V'T'} \right) + \kappa_h \nabla_h^2 \overline{T} \right) dz \right] \\ & - \rho C_P \left( \hat{T} - \overline{T}(z_0) \right) \overline{w}(z_0) - \rho C_P \left[ \left( \overline{w'T'} \right)_0 - \left( \overline{w'T'} \right)_{z_0} \right] \\ & - \rho C_P \kappa_v \left. \frac{\partial \overline{T}}{\partial z} \right|_{z_0} \end{aligned} \tag{A10}$$

(A10) is the Equation (2) in the body text of the paper. Integrating the (A10) in time, estimate of each heat term can be obtained.

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
