# Peer review of "Upper-Ocean Processes Controlling the Near-Surface Temperature in the Western Gulf of Mexico from a Multidecadal Numerical Simulation"

_2673-4834, doi:10.3390/earth3020030_

Round 1
Reviewer 1 Report
The manuscript explores the impact of upper ocean processes on the near surface temperature in the western Gulf of Mexico. Eddy kinetic energy, heat fluxes, heat advection, and other dynamic characteristics are analyzed for the study region. I find the manuscript to be interesting. Although the direct results are only applicable to the Gulf of Mexico, the analysis could be applied to any number of locations as long as data are available. Overall, the manuscript is well written, it is easy to read, and well structured. The figures are well done and of good quality.
My only comment is that there need to be some changes in the figures. In particular, in Figures 1-3, I propose to add the differences between the models and the observations or reanalysis data. Furthermore, Figures 10 to 12 misleads due to the small vertical heat advection values, it is a confusing spaghetti diagram. Ι suggest changing the format of the figures, a format where the differences between the parameters can be seen is preferable. Finally, the capture of Table 2 is too extended, it should be shortened, the equations should be removed.
I can reconsider this article for publication after minor revision.
Author Response
Dear Reviewer,
We have addressed all your concerns in the attached response letter. We really appreciate your comments which improve the paper.
Sincerely,
Yangxing and co-authors

Reviewer 2 Report
The manuscript well describes Upper-Ocean Processes Controlling the Near Surface Temperature in the Western Gulf of Mexico. In this present form the work has an high quality but some minor revision must be performed after going ahead in order to imporve the quality of the work.
The introduction can be improve by considering to shortly describe also the effect of climate on land ecosystems as a general background before focusing on your topics. Earth journal try to focus on different earth system component. Here it is some good paper that can be insert into the introduction section.
- https://doi.org/10.3390/cli9030047
- https://doi.org/10.3390/rs13010044
After this section i adivise you to write a short section called study area in which you describe according the topic of your work the modelled area to facilitate the reader.
In the methods better describe the tools adopted and the input parameters not only the equation.
The rest part is fine in this present form according the present reviewer.
Author Response
Dear Reviewer,
Thank you for your great effort in this paper. We have addressed all your concerns in the attached response letter.
Sincerely,
Yangxing and co-authors
